# Tumors defective in homologous recombination rely on oxidative metabolism: relevance to treatments with PARP inhibitors

Álvaro Lahiguera[1,2], Petra Hyroššová[3], Agnès Figueras[1,2], Diana Garzón[1,2], Roger Moreno[1,2], Vanessa Soto-Cerrato[4], Iain McNeish[5], Violeta Serra[6,7], Conxi Lazaro[2,7,8], Pilar Barretina[9], Joan Brunet[7,8,10,11], Javier Menéndez[12,13], Xavier Matias-Guiu[2,7,14], August Vidal[2,4,14,15], Alberto Villanueva[1,2,15], Barbie Taylor-Harding[16], Hisashi Tanaka[16], Sandra Orsulic[17], Alexandra Junza[18,19], Oscar Yanes[18,19], Cristina Muñoz-Pinedo[20], Luís Palomero[1,2], Miquel Àngel Pujana[1,2], José Carlos Perales[3] & Francesc Viñals[1,2,3,*]

## Abstract

Mitochondrial metabolism and the generation of reactive oxygen species (ROS) contribute to the acquisition of DNA mutations and genomic instability in cancer. How genomic instability influences the metabolic capacity of cancer cells is nevertheless poorly understood. Here, we show that homologous recombination-defective (HRD) cancers rely on oxidative metabolism to supply $NAD^+$ and ATP for poly(ADP-ribose) polymerase (PARP)-dependent DNA repair mechanisms. Studies in breast and ovarian cancer HRD models depict a metabolic shift that includes enhanced expression of the oxidative phosphorylation (OXPHOS) pathway and its key components and a decline in the glycolytic Warburg phenotype. Hence, HRD cells are more sensitive to metformin and $NAD^+$ concentration changes. On the other hand, shifting from an OXPHOS to a highly glycolytic metabolism interferes with the sensitivity to PARP inhibitors (PARPi) in these HRD cells. This feature is associated with a weak response to PARP inhibition in patient-derived xenografts, emerging as a new mechanism to determine PARPi sensitivity. This study shows a mechanistic link between two major cancer hallmarks, which in turn suggests novel possibilities for specifically treating HRD cancers with OXPHOS inhibitors.

**Keywords** BCRA; cancer metabolism; metformin; OXPHOS; PARP inhibitors
**Subject Categories** Cancer; Metabolism

## Introduction

Genomic instability and deregulation of cellular energetics are two of the hallmarks of cancer cells, as re-defined by Hanahan and Weinberg (2011). Genomic instability arises when mechanisms that protect genomes from alterations in DNA are overcome. Consequently, as

1 Program Against Cancer Therapeutic Resistance (ProCURE), Institut Català d'Oncologia, Hospital Duran i Reynals, L'Hospitalet de Llobregat, Barcelona, Spain
2 Oncobell Program, Institut d'Investigació Biomèdica de Bellvitge (IDIBELL), L'Hospitalet de Llobregat, Barcelona, Spain
3 Departament de Ciències Fisiològiques, Universitat de Barcelona, Barcelona, Spain
4 Departament de Patologia i Terapèutica Experimental, Universitat de Barcelona, Barcelona, Spain
5 Department of Surgery and Cancer, Imperial College, London, UK
6 Experimental Therapeutics Group, Vall d'Hebron Institute of Oncology, Barcelona, Spain
7 CIBERONC, Instituto de Salud Carlos III, Madrid, Spain
8 Hereditary Cancer Program, Institut Català d'Oncologia, Hospital Duran i Reynals, L'Hospitalet de Llobregat, Barcelona, Spain
9 Medical Oncology Department, Institut Català d'Oncologia, IDIBGI, Girona, Spain
10 Hereditary Cancer Program, Institut Català d'Oncologia, IDIBGI, Girona, Spain
11 Medical Sciences Department, School of Medicine, University of Girona, Girona, Spain
12 Program against Cancer Therapeutic Resistance (ProCURE), Metabolism and Cancer Group, Catalan Institute of Oncology, Girona, Spain
13 Girona Biomedical Research Institute (IDIBGI), Girona, Spain
14 Servei d'Anatomia Patològica, Hospital Universitari de Bellvitge, L'Hospitalet de Llobregat, Barcelona, Spain
15 Xenopat, Carrer de la Feixa Llarga S/N, L'Hospitalet de Llobregat, Barcelona, Spain
16 Womens Cancer Program, Cedars-Sinai Medical Center, Los Angeles, CA, USA
17 David Geffen School of Medicine at UCLA, Los Angeles, CA, USA
18 Metabolomics Platform, Department of Electronic Engineering (DEEEA), Universitat Rovira i Virgili, Tarragona, Spain
19 Biomedical Research Centre in Diabetes and Associated Metabolic Disorders (CIBERDEM), Madrid, Spain
20 Cell Death Regulation Group, Oncobell Program, Bellvitge Biomedical Research Institute (IDIBELL), L'Hospitalet de Llobregat, Barcelona, Spain
*Corresponding author. Tel: +34 93 2607344; E-mails: fvinyals@iconcologia.net; fvinals@ub.edu

mutation rates increase, cells can adapt better to changes in the tumor environment, response to therapies, etc. An increase in genomic instability could be caused by alterations in the DNA damage response (DDR), including the mechanisms responsible for (i) detecting DNA damage, (ii) DNA repair, and/or (iii) detoxification (Jackson & Bartek, 2009). The DNA repair machinery corrects, among other things, double- and single-strand DNA breaks (DSBs and SSBs, respectively). DSBs are repaired by processes such as homologous recombination (HR) and non-homologous end joining (NHEJ), and SSBs are repaired by base excision repair (BER) and nucleotide excision repair (NER) mechanisms. BRCA1 and BRCA2 proteins play a key role in HR, and mutations in these genes cause defective HR and increased genomic instability (Roy et al, 2011). Poly(ADP-ribose) polymerase (PARP) proteins are DNA damage sensors because they can bind to breaks and nicks in DNA and transduce signals to the DNA repair machinery by attaching branched poly(ADP-ribose) (PAR) chains to various proteins (Krishnakumar & Kraus, 2010; Gupte et al, 2017; Lord & Ashworth, 2017). Cancer cells with defective HR depend on alternative DNA repair pathways, such as BER and NHEJ, both of which are promoted by PARP enzymes. This is the basis of the clinical utility of PARP inhibitors in HR-deficient cases (Lord & Ashworth, 2017).

Tumoral cells must maintain a constant supply of nutrients and energy to sustain the growth and cell division that characterize cancer (Boroughs & DeBerardinis, 2015). In order to do so, a large proportion of tumoral cells switch their metabolism to aerobic glycolysis (the Warburg effect), a preference for increased glycolysis regardless of the presence of oxygen (Hsu & Sabatini, 2008; Vander Heiden et al, 2009; Ward & Thompson, 2012). Thus, glycolytic intermediates expand to cope with biosynthetic pathways, such as those involved in the synthesis of nucleotides and amino acids (Hsu & Sabatini, 2008; Vander Heiden et al, 2009; Ward & Thompson, 2012; Hay, 2016). However, a preference for aerobic glycolysis does not always involve reduced respiratory capacity. In fact, tumoral cells depend on a continuing supply of TCA cycle intermediates, such as those serving as direct precursors of lipids, and those providing carbons and cofactors in the nucleotide and amino acid synthesis pathways (Vyas et al, 2016). Thus, depending on the tumoral cells' intrinsic capacity (oncogene profile, metabolic program, proliferative state, etc), the characteristics of the tumor environment (e.g., with or without hypoxia, and metabolic substrates produced by the tumoral stroma), and the effects of the cancer treatment (e.g., presence of drugs and ROS generated by the treatment), the metabolism of cancer cells can change and readapt. Examples of this are cancer cells subjected to chronically low in vitro glucose levels, which increase oxidative phosphorylation (OXPHOS) to maintain growth (Birsoy et al, 2014), chemotherapy-resistant breast tumoral cells, which have a higher mitochondrial oxidative metabolic rate than do sensitive cells (Janzer et al, 2014), and antiangiogenic treatment in breast and lung cancer models that induce a shift from a largely glycolytic metabolism to a more oxidative one (Navarro et al, 2016).

The capacity to detect and repair DNA damage and the metabolic characteristics of a cell are coordinated (Berkers et al, 2013; Imai & Guarente, 2014; Shimizu et al, 2014). Different repair mechanisms rely on specific metabolites, which may, in turn, limit the capacity of cancer cells to cope with different chemotherapies. PARP enzymes directly depend on the cellular metabolic state, particularly on $NAD^+$ levels, for their functions (Krishnakumar & Kraus, 2010;

Gupte et al, 2017); hence, a direct role for PARP proteins linking DNA damage and metabolic capacities in different situations has been proposed (Bai et al, 2007; Sakamaki et al, 2009; Brace et al, 2016). Acute DNA damage inhibits glutamine metabolism to control cell cycle arrest through SIRT4, a family of proteins that also depend on $NAD^+$ for their activity (Jeong et al, 2013). Meanwhile, DDR upregulates the pentose phosphate pathway and glycolysis to promote nucleotide synthesis and an open global chromatin structure (Cosentino et al, 2011; Liu et al, 2015; Efimova et al, 2016). However, the metabolic requirements of BRCA-dependent HR DNA repair mechanisms remain unknown. In this study, analyses of cancer cell models, patient-derived xenografts (PDXs), and human cancer samples reveal novel mechanisms linking HR defects to increased oxidative metabolism. Triggering oxidative metabolism in HR-defective (HRD) cancer cells emerges as a fundamental characteristic of cancer survival due to its contribution to DNA damage repair, a feature set that highlights additional cancer vulnerabilities, as it emerges as a novel player in determining sensitivity to PARP inhibitors.

## Results

### HR deficiency is linked to enhanced OXPHOS gene expression in breast and ovarian cancer

To identify metabolic pathways that may be associated with genomic instability due to HR deficiency, we integrated somatic mutational profiles and gene expression data from the breast and ovarian cancer studies of TCGA (Cancer Genome Atlas Research Network, 2011; Cancer Genome Atlas Network, 2012). Tumors were classified according to the presence (S3+) or absence (S3−) of mutational signature 3, originally shown to be characteristic of cases with HR defects and BRCA1/2 mutations (Alexandrov et al, 2013). Next, an association with changes in gene expression of metabolic pathways sets was analyzed using the Gene Set Enrichment Analysis tool (Subramanian et al, 2005) and pathway annotations from the Kyoto Encyclopedia of Genes and Genomes (KEGG) (Kanehisa et al, 2017). These analyses revealed that OXPHOS and four additional mitochondria-related metabolic sets (Table EV1) are positively associated with the presence of the S3 signature (S3+), that is, with HR defects in breast tumors (gene set expression analysis [GSEA] nominal $P = 0.019$, Fig 1A, top panel). Moreover, we also complied BRCA1/2 mutation status of TCGA breast cancer datasets from different sources (cBioPortal and TCGA accessible data) (Kraya et al, 2019; Yost et al, 2019) and further analyzed the association with the KEGG oxidative phosphorylation pathway. In multivariate regression analyses including age at diagnosis and tumor stage, overexpression of this pathway was also detected significantly associated with BRCA1/2 mutations ($P = 0.013$, Fig 1A, middle panel). Consistent with this observation, the single-sample GSEA (ssGSEA) scores of the KEGG OXPHOS gene set were found to be negatively correlated (Pearson's correlation coefficients [PCCs] ≤ −0.15, $P < 0.001$) with both BRCA1 and BRCA2 expression levels (Fig 1A, bottom panels). Subsequently, a similar metabolic association was observed with high-grade serous ovarian tumors positive for the mutational signature 3: higher OXPHOS gene expression in S3+, HR-defective tumors (false-discovery rate [FDR]-adjusted $P = 0.025$, Fig 1B, top

panel) and with *BRCA1/2* mutation status of TCGA ovarian cancer datasets ($P < 0.001$, Fig 1B, middle panel). The OXPHOS gene set was also found to be negatively correlated (PCCs $\leq -0.18$, $P < 0.01$) with *BRCA1* and *BRCA2* expression (Fig 1B, bottom panels).

Homologous recombination-defective tumors present a better response (are more sensitive) to some chemotherapy treatments, such as cisplatin in ovarian cancers (Rigakos & Razis, 2012; Safra *et al*, 2014). To evaluate the link between OXPHOS metabolic activity and response to chemotherapy in HRD cancers, data from the Cambridge Translational Cancer Research Ovarian Study 01 (https://www.ncbi.nlm.nih.gov/pubmed/18068629) were analyzed. The level of expression of the OXPHOS gene signature was significantly higher in pre-treated ovarian tumors that subsequently showed higher sensitivity to carboplatin (Fig 1C, top panel). Importantly, analogous differences were found for an HRD signature (Fig 1C, bottom panel; https://www.nature.com/articles/ncomms4361). Indeed, an association between the OXPHOS gene signature in ovarian tumors and clinical outcome was evident from higher survival probability observed in patients displaying high OXPHOS expression (Appendix Fig S1A).

Next, to better delineate the association in ovarian cancer, we analyzed a retrospective series of high-grade serous epithelial ovarian carcinomas without or with mutations in *BRCA2*. Immunohistochemical analyses revealed downregulation of the monocarboxylate transporter MCT4 (which is responsible for lactate extrusion in glycolytic cells; Fig 1D), and overexpression of the OXPHOS complex I respiratory chain NDUFV2 subunit (one of the genes that were increased in the HR-defective-associated S3+ signature; Fig 1E) in *BRCA2*-mutated tumors relative to wild-type high-grade serous cases. The analysis of a second independent series of high-grade serous epithelial ovarian carcinomas (with and without mutations in *BRCA2)* also showed identical changes in MCT4 and NDUFV2 protein expression by Western blot (Fig 1F). We also measured the proliferative capacity of WT or *BRCA2*-mutated human ovarian high-grade serous tumors by Ki67 staining, but we did not observe changes in tumoral cell proliferation rates associated with the HRD phenotype (Appendix Fig S1B).

Collectively, these results from two cancer types with frequent inactivation of HR indicate an association between this cancer hallmark and enhanced OXPHOS.

## Enhanced oxygen consumption in genomic instable ovarian cancer cells

To functionally corroborate the above gene/protein expression associations, oxidative metabolism was evaluated in cancer cell models characterized by defective HR and high or low genomic instability using their genomic analysis, olaparib sensitivity (high sensitivity implies high genomic instability), and phospho(Ser129) histone H2A.X levels (marker of double-strand DNA breaks) as surrogate indicators (Appendix Figs S2–S4). These models included (i) murine ID8 ovarian cancer cells wild-type, *Trp53*-deficient (less sensitive to olaparib), or *Trp53* and *Brca2*-deficient (olaparib-sensitive) (Walton *et al*, 2016); (ii) SKOV-3-SC (control cells infected with an empty vector, *TP53* mutant but *BRCA1/2* wild-type, less sensitive to olaparib) or their SKOV-3-*BRCA2*$^{-/-}$ 1.2 counterpart (olaparib-sensitive); (iii) SKOV-3 cells (less sensitive to olaparib) or their cisplatin-resistant counterpart SKOV-3-R (with high genomic

instability and olaparib-sensitive); (iv) murine ovarian cancer cells obtained from tumors generated in mice with combinations of genetic alterations *Trp53*$^{-/-}$, *myc*, and *Akt* (less sensitive to olaparib) or *Trp53*$^{-/-}$, *Brca1*$^{-/-}$, *myc*, and *Akt* (olaparib-sensitive) (Xing & Orsulic, 2006); and (v) murine *Trp53*-deficient MEFs (less sensitive to olaparib) or double *Trp53/Tsc2*-deficient MEFs (with higher genomic instability and more olaparib-sensitive) (Zhang *et al*, 2003; Sun *et al*, 2014). Measurements of oxygen consumption in these cells in real time using a high-resolution oxygraph showed that genomic instable/HRD cells consumed oxygen at higher rates (Figs 2A and EV1). The lowest rate of consumption was found in ID8 control cells (i.e., wild-type for both *Trp53* and *Brca2*; $43 \pm 4.3$ pmol/s), whereas higher levels were observed in *Trp53*-deleted ($62 \pm 4.9$ pmol/s) and, subsequently, in *Trp53/Brca2*-deleted ID8 cells ($79 \pm 6.1$ pmol/s; Fig 2A). Similarly, SKOV-3-*BRCA2*$^{-/-}$ 1.2 cells consumed more oxygen than SKOV-3 SC control cells ($78.1 \pm 7.3$ and $53.0 \pm 3.2$ pmol/s, respectively, Fig EV1A), SKOV-3-R consumed more oxygen than SKOV-3 cells ($58 \pm 5$ and $39 \pm 4$ pmol/s, respectively, Fig EV1B), murine ovarian cancer *Trp53*$^{-/-}$, *Brca1*$^{-/-}$, *myc*, and *Akt* cells consumed more oxygen than *Trp53*$^{-/-}$, *myc*, and *Akt* cells ($67.5 \pm 4.0$ and $54.0 \pm 2.8$ pmol/s, respectively, Fig EV1C), and double *Trp53/Tsc2*-deleted MEFs consumed more oxygen than *Trp53*-deficient MEFs ($66.3 \pm 4.2$ and $33.4 \pm 4.0$ pmol/s, respectively, Fig EV1D), linking increased oxygen consumption to HR defects and greater genomic instability. In turn, cell lines characterized by a high microsatellite instability (MSI-H) phenotype relative to MSI stable and low microsatellite instability cancer cell lines presented a higher expression of a signature defining mitochondrial respiratory chain complex I (Appendix Fig S5).

## Enhanced OXPHOS metabolism is coupled to decreased glycolysis

The main site of oxygen consumption in the cell is the respiratory chain of the mitochondria. Increased oxygen consumption may be due to higher respiratory capacity and/or a more efficient flux of substrates to the mitochondria. Analysis of mitochondrial mass with a mitochondria specific fluorescence dye (non-potential-dependent MitoTracker) in ID8 cell models showed small but no significant differences (Appendix Fig S6A). Expression of VDAC, an anion channel in the internal mitochondrial membrane, was similarly indicative of unaffected mitochondria mass in these models (Appendix Fig S6B). These results suggest that the higher rates of oxygen consumption observed in the cell sets with HRD were not caused by alteration in mitochondria turnover or biogenesis, but instead by changes in the flux of substrates toward oxidation.

As glucose is finely regulated to either feed the OXPHOS or lactate production, we decided to analyze glucose metabolism in ID8 cell models. The *Trp53/Brca2*-deleted cells consumed significantly less glucose (Fig EV2A) and produced less lactate (Fig 2B) than *Trp53*-deleted cells. Overall, these results indicate a drop in the glycolytic capacity of HRD cells. To confirm this, we measured the glycolytic flux by evaluating $^3H_2O$ production from $^3H$-C$_5$-Glucose at the enolase step of glycolysis. Glycolytic flux measured in *Trp53/Brca2*-deleted cells was 18% lower than in *Trp53*-deleted cells (Fig 2C). A marked drop in key glycolytic enzymes (a 50% reduction in hexokinase and a 53% reduction in lactate dehydrogenase mRNAs; Fig EV2B) and transporters (a 55% reduction in GLUT1

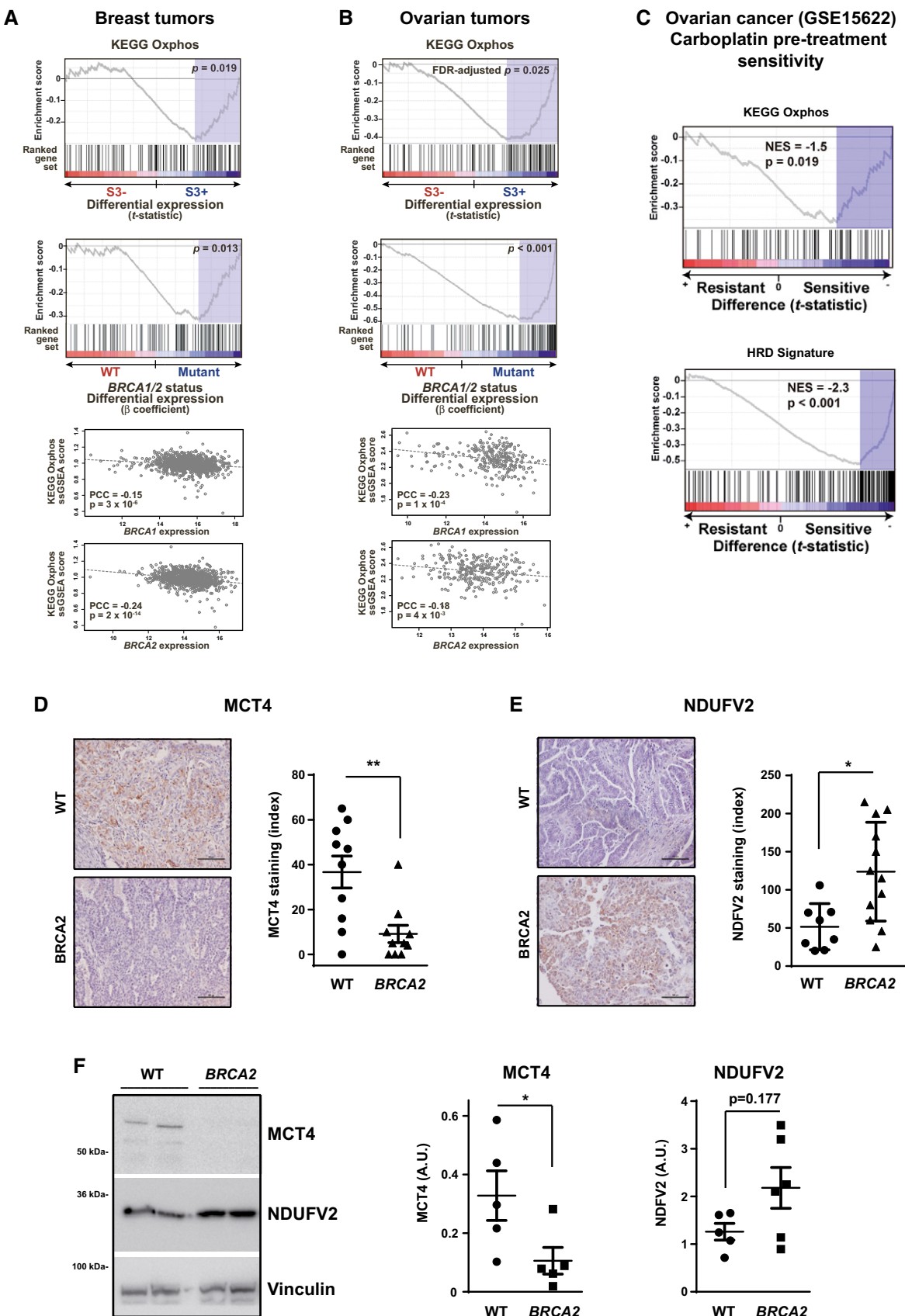

**Figure 1.**

**Figure 1. HR defects are associated with OXPHOS gene overexpression.**

A, B GSEA results regarding the association between OXPHOS gene set overexpression and positivity for mutational signature 3 (associated with HR defects) in TCGA breast cancers (A) and TCGA ovarian cancer data (B). Top panel, enrichment score, gene ranking (based on the *t*-statistic of the expression differences between negative (S3−) and positive (S3+) tumors), and false-discovery rate (FDR)-adjusted *P* values are shown. Middle panel shows similar GSEA results using as metric the β coefficient of differential expression between BRCA1/2 wild-type and mutant tumors, including the covariates of age at diagnosis and tumor stage. Bottom panels, scatter plots showing the correlations (Pearson's correlation coefficients and *P* values) between the ssGSEA scores of the OXPHOS gene set and the *BRCA1* (top) and *BRCA2* (bottom) somatic gene expression values.

C GSEA results of KEGG OXPHOS (top panel) and HRD (bottom panel) signature score comparisons between carboplatin-resistant (left) and carboplatin-sensitive (right) ovarian tumors, using pre-treatment expression data (GSE15622 data). The normalized enrichment scores (NESs) and corresponding *P* values are indicated. The NES is negative because the comparison is between resistant and sensitive tumors, so negative values mean that expression is higher in the second term (i.e., sensitive tumors).

D Left panel, MCT4 staining of wild-type and *BRCA2*-mutated high-grade serous ovarian tumors. Scale bars, 100 μm. Right panel, quantification of MCT4 levels using the multiplicative index in tumor tissue sections from wild-type (*n* = 10) and *BRCA2*-mutated (*n* = 8) high-grade serous tumors. Error bars indicate the SEM. \*\**P* = 0.0055 (two-tailed unpaired Mann–Whitney *U*-test).

E Left panel, NDUFV2 staining of wild-type and *BRCA2*-mutated high-grade serous ovarian tumors. Scale bars, 100 μm. Right panel, quantification of NDUFV2 levels using the multiplicative index in tumor tissue sections from wild-type (*n* = 8) and *BRCA2*-mutated (*n* = 12) high-grade serous tumors. Error bars indicate the SEM. \**P* = 0.0101 (two-tailed unpaired Mann–Whitney *U*-test).

F Samples from wild-type (WT, *n* = 5) and *BRCA2*-mutated (*n* = 6) high-grade serous ovarian tumors were lysed and immunoblotted using the indicated antibodies. Left panel: A representative blot of two independent tumors is shown. Right panel: Bars show the relative immunoreactivity of MCT4 or NDUFV2 proteins normalized with respect to vinculin in samples from the left. Error bars indicate the SEM. Statistical significance of two-tailed unpaired Mann–Whitney *U*-tests, \**P* = 0.0317.

Source data are available online for this figure.

protein, the main glucose transporter; Fig 2D) paralleled changes in glycolytic flux. Moreover, in comparison with *Trp53*-deleted cells, *Trp53*/*Brca2*-deleted cells revealed higher (∼ 3.7-fold) TIGAR protein expression, an inhibitor of the glycolysis (Bensaad *et al*, 2006; Ko *et al*, 2016) (Fig 2D). These results demonstrate that HR defects reduced glycolytic flux.

Next, we set out to establish the metabolic regulatory axis in *Trp53*/*Brca2*-deleted cells by examining key control nodes, such as pyruvate dehydrogenase (PDH), the mitochondrial enzyme responsible for converting pyruvate to acetyl-CoA, a step prior to its complete oxidation in the tricarboxylic acid (TCA) cycle and subject to multiple mechanisms of regulation. One of these mechanisms of regulation involves the phosphorylation of PDH at the E1 subunit which renders the enzyme inactive, effectively diverting the flux of pyruvate away from the mitochondria (Fig 2F). Extracts from *Trp53*/*Brca2*-deleted cells had 58% lower levels of phosphorylated PDH than those of *Trp53*-deleted cells (Fig 2E). As phosphorylation blocks the activity of PDH, this result indicates that the pyruvate pools of *Trp53*/*Brca2*-deleted cells tend to be preferentially distributed into the mitochondria, and consequently, they have a greater capacity to fully oxidize glucose into $CO_2$ and water. Similar changes in phosphorylated PDH were measured in murine ovarian cancer $Trp53^{-/-}$, $Brca1^{-/-}$, *myc*, and *Akt* cells and in double *Trp53*/*Tsc2*-deleted MEFs as compared to $Trp53^{-/-}$, *myc*, and *Akt* cells and *Trp53*-deleted MEFs (Fig EV2C and D). *Trp53*/*Brca2*-deleted cells also expressed more phosphorylated (inactive) acetyl-CoA carboxylase (ACC; Fig 2E), which is a node for the regulation of fatty acid synthesis and oxidation. This observation suggested that *Trp53*/*Brca2*-deleted cells could also oxidize more fatty acids than *Trp53*-deleted cells. To address this hypothesis, cells were incubated in the presence of different metabolic substrates, and oxygen consumption was measured in real time. These experiments demonstrated that glucose was the only substrate capable of increasing oxygen consumption specifically in *Trp53*/*Brca2*-deleted cells and ruled out enhanced consumption of fatty acids or glutamine (Fig EV2E). Finally, targeted metabolomic enrichment analysis using $C^{13}$-U-Glucose as a substrate was used to trace TCA cycle dynamics in

*Trp53*/*Brca2*-deleted and *Trp53*-deleted cells (Fig 2F). M+3 and m+4 isotopologs of fumarate, aspartate (equilibrating with the oxaloacetate pool), and glutamate (equilibrating with the α-ketoglutarate pool) were enriched significantly in *Trp53*/*Brca2*-deleted cells, illustrating increased entry, and cycling, of fully labeled pyruvate, and is consistent with enrichment of citrate m+5 in these same cells. These data are in agreement with the oxidative profile shown for these cells and their enhanced oxygen respiratory rates.

Collectively, these data indicated that OXPHOS is increased in *Trp53*/*Brca2*-deleted cells secondary to, at least, increased contribution of glucose to mitochondrial metabolism (Fig 2F).

## OXPHOS metabolism sensitizes genomically instable cells to metformin

Enhanced OXPHOS in HR-deficient cells may foresee novel vulnerabilities in the corresponding tumors. Indeed, *Trp53*/*Brca2*-deleted cells were more sensitive than *Trp53*-deleted cells to a blockade of respiratory complex I (NADH-ubiquinone oxidoreductase) using metformin (half-maximal inhibitory concentration [$IC_{50}$] of $5.2 \pm 0.5$ and $8.2 \pm 0.7$ mM, respectively; *P* = 0.0006, Fig 3A) (El-Mir *et al*, 2000; Owen *et al*, 2000; Wheaton *et al*, 2014). To confirm that the effect of metformin was dependent on its effect on the mitochondrial respiratory chain, cells were incubated with rotenone, a more potent inhibitor of complex I. Again, *Trp53*/*Brca2*-deleted cells were 67% more sensitive than *Trp53*-deleted cells to the drug ($IC_{50}$ of $16.3 \pm 1.5$ and $48.9 \pm 8.2$ nM, respectively; *P* = 0.028, Fig 3B). *Trp53*/*Brca2*-deleted were also more sensitive to inhibition of complex V (ATP synthase) by oligomycin than *Trp53*-deleted cells ($IC_{50}$ of $0.53 \pm 0.10$ and $1.36 \pm 0.26$ nM, respectively; *P* = 0.028, Fig 3B), suggesting that multiple bioenergetic components are implicated in the differential sensitivity of *Trp53*/*Brca2*-deleted cells to metformin.

To identify the reasons for the differential effect of metformin, we studied the effect of this drug on cell proliferation and cell death. Metformin affected cell viability disrupting the cell cycle of Trp53-deleted cells and Trp53/Brca2-deleted cells (Fig EV3A), with

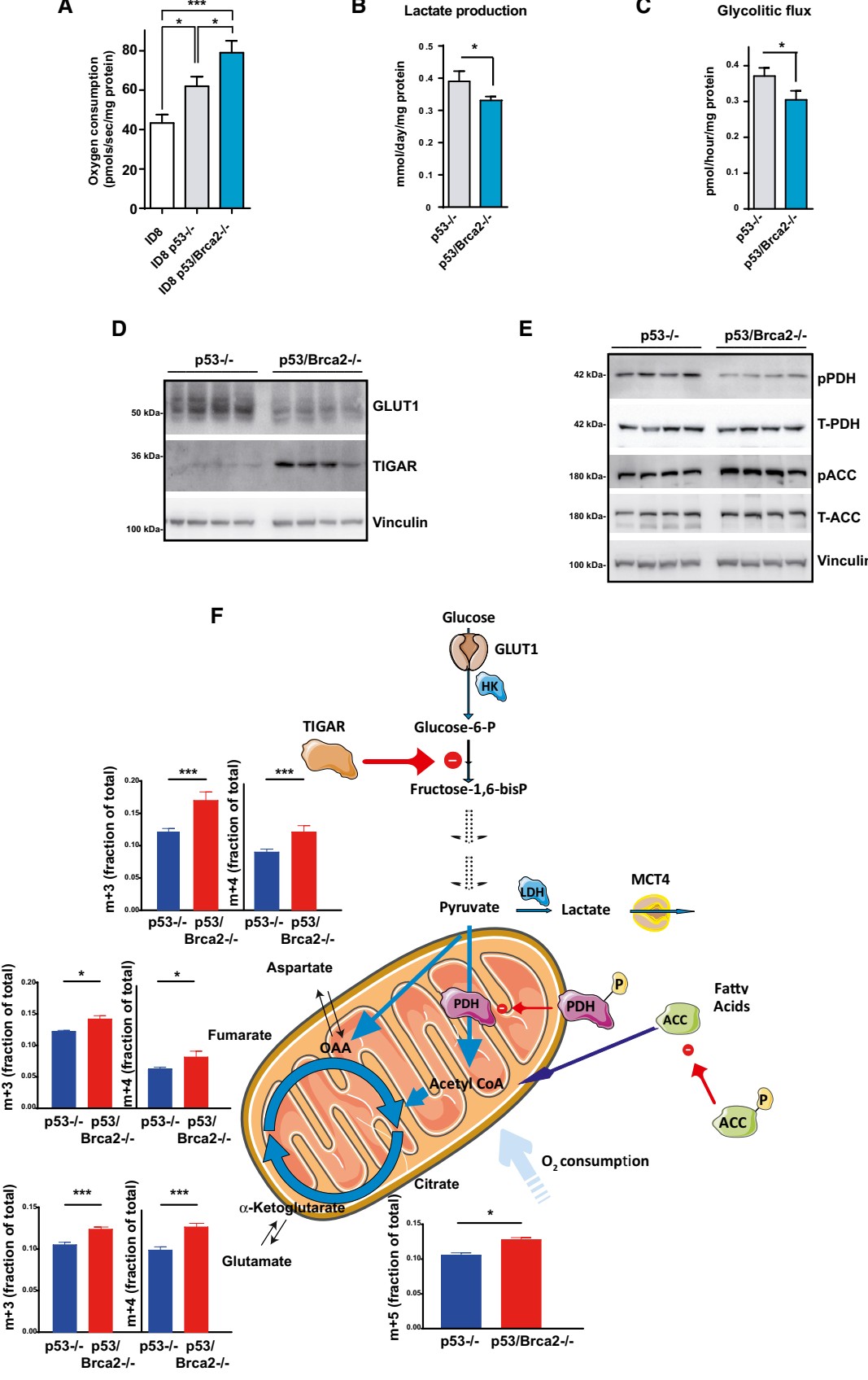

**Figure 2.**

**Figure 2.  Genomic instable cell models show high oxygen consumption and decreased glycolytic flux.**

A  Oxygen consumption rates measured using an Oxygraph-2K in ID8 control ($n = 10$), ID8 *Trp53*-deleted ($n = 18$), and double *Trp53/Brca2*-deleted ($n = 17$) mouse ovarian tumor cells. The bars indicate the mean and standard error (SEM). Statistical significance of two-tailed unpaired Mann–Whitney *U*-tests: ID8 control versus ID8 *Trp53*-deleted, *$P = 0.0266$; ID8 control versus double *Trp53/Brca2*-deleted, ***$P = 0.0009$; ID8 *Trp53*-deleted versus double *Trp53/Brca2*-deleted, *$P = 0.0204$.

B  Lactate production (mmol) was measured in DMEM from ID8 *Trp53*-deleted ($n = 4, 0.36, 0.42, 0.37$, and $0.38$) and *Trp53/Brca2*-deleted ($n = 4, 0.33, 0.31, 0.33$, and $0.33$) ovarian tumor cells grown for 24 h. Data were normalized with respect to protein content. Error bars indicate the SEM. Statistical significance of two-tailed unpaired Mann–Whitney *U*-tests, *$P = 0.0236$.

C  Glycolytic flux was measured in ID8 *Trp53*-deleted ($n = 4, 0.346, 0.361, 0.377$, and $0.400$) and *Trp53/Brca2*-deleted ($n = 3, 0.276, 0.324$, and $0.314$) ovarian tumor cells grown for 24 h in DMEM complete medium and in the presence $^3$H-glucose for the final 60 min. Measurements (in pmol $^3$H $H_2O$ produced) were normalized relative to protein content. Error bars indicate the SEM. Statistical significance of two-tailed unpaired Mann–Whitney *U*-tests, *$P = 0.0153$.

D  ID8 *Trp53*-deleted ($n = 4$) and *Trp53/Brca2*-deleted ($n = 4$) ovarian tumor cells were grown in complete DMEM for 3 days. Cells were lysed and immunoblotted using the indicated antibodies.

E  ID8 *Trp53*-deleted ($n = 4$) and *Trp53/Brca2*-deleted ($n = 4$) ovarian tumor cells were grown in complete DMEM for 3 days. Cells were lysed and immunoblotted using the indicated antibodies.

F  Targeted metabolomic enrichment analysis using $C^{13}$-U-Glucose as substrate. M+3 (left panels) isotopolog enrichment and m+4 (right panels) isotopolog enrichment of fumarate, aspartate (equilibrating with the oxaloacetate pool), and glutamate (equilibrating with the α-ketoglutarate pool) are shown to illustrate TCA cycle entry of pyruvate carbons, and cycling. Citrate m+5 enrichment is also shown as a measure of the capacity of the mitochondria to oxidize fully labeled pyruvate. The mean of four replicates for each cell line is shown. Error bars indicate the SEM. Statistical significance of citrate m+5 by two-tailed unpaired Mann–Whitney *U*-test, *Trp53*-deleted versus *Trp53/Brca2*-deleted cells *$P = 0.0286$. Statistical significance of one-way ANOVA, Holm–Sidak's multiple comparisons test *Trp53*-deleted versus *Trp53/Brca2*-deleted cells: Glutamate, m+3 ***$P < 0.0001$, m+4 ***$P < 0.0001$; fumarate, m+3 *$P = 0.0033$, m+4 *$P = 0.0051$; aspartate, m+3 ***$P < 0.0001$, m+4 ***$P < 0.0001$.

Source data are available online for this figure.

no increase in apoptosis (Fig EV3B); in contrast, administering olaparib led to a substantial increase in the number of apoptotic cells (Fig EV3B). High OXPHOS metabolism is usually associated with an increase in ROS. We did not detect differences in mitochondrial ROS basal levels between *Trp53/Brca2*-deleted cells and *Trp53*-deleted cells, although incubation with metformin caused a higher increase in mitochondrial ROS in *Trp53/Brca2*-deleted cells than in *Trp53*-deleted cells, as assessed by surrogate mitoSOX labeling (Appendix Fig S7).

Cancer cells with defective HR are completely dependent on alternative DNA repair pathways promoted by $NAD^+$ and ATP-consuming PARP enzymes, and alterations in the activity of these enzymes will affect cell viability. In fact, *Trp53/Brca2*-deleted cells presented high phospho-histone H2A.X levels when treated with metformin (Figs 3C and EV4). To determine whether oxidative metabolism changes observed in *Trp53/Brca2*-deleted cells correlated with PARP activity, first we measured PARP1 protein levels. *Trp53/Brca2*-deleted cells presented lower PARP1 levels compared with *Trp53*-deleted cells (Fig 3D). Next, the levels of poly-ADP-ribosylated (PAR) proteins were measured. PARP activity in basal conditions was 25% lower in *Trp53/Brca2*-deleted cells than in *Trp53*-deleted cells (Fig 3E), in agreement with PARP1 protein levels. As expected, incubation with olaparib completely blocked PARP activity in both cell types. Metformin significantly affected PAR levels in both cell types, but the effect of metformin incubation on PARP activity was stronger in *Trp53/Brca2*-deleted cells (63% versus 40% reduction, Fig 3E). In aggregate, HRD cells maintain a very low level of PARP activity in the presence of metformin, like those observed after PARP inhibition with olaparib. These results suggested that OXPHOS is involved in maintaining PARP activity. To confirm this, we increased pyruvate flux to the mitochondria by incubation with dichloroacetate (DCA), an inhibitor of PDH kinase, therefore releasing the inhibition of PDH sustained by phosphorylation by this kinase (Stacpoole *et al*, 2008). As expected, DCA incubation caused an increase in oxygen consumption (16%), but more importantly, this increase in OXPHOS was associated with an increase in PARP

activity (Fig 3F). In all, these results confirmed that OXPHOS is involved in maintaining PARP activity, essential for cell survival and proliferation in HRD cells, and partially explaining the dramatic effect of metformin in these cells.

## Oxidative metabolism is necessary to maintain high ATP levels and $NAD^+$/NADH ratio

The above results depict a biological association between enhanced OXPHOS and defective HR cancer cells that serves to maintain a high level of activity of PARP enzymes. However, it was unclear how increased levels of OXPHOS were linked to PARP activity. To investigate this matter, we analyzed the metabolic consequences of metformin action in these cells, by first measuring the fluxes of glucose carbon into metabolites of the TCA cycle and its branching pathways (Fig 4A). As expected, the inhibition of complex I by metformin was met with a marked reduction in fluxes to TCA cycle intermediaries as shown by reduced contribution (ratio of fully labeled and unlabeled isotopologs) of carbons from $C^{13}$-U-glucose analyzed by targeted metabolomics (Fig 4A). However, carbon flux and metabolite pools were unsensitive to the presence of Brca2, as metformin affected both cell lines similarly.

Next, we studied energy charge (total ATP levels) in these cells. *Trp53/Brca2*-deleted cells presented high ATP levels (74% higher) relative to what was measured in *Trp53*-deleted cells (Fig 4B). However, ATP levels after 48-h incubation with metformin dropped by 42% in *Trp53/Brca2*-deleted cells compared with a milder reduction of 11% in *Trp53*-deleted cells (Fig 4B), suggesting that OXPHOS metabolism is consequential for maintaining high ATP levels in HRD cells.

We subsequently analyzed the impact of altered redox equilibrium in HR-deficient cells. PARP enzymes depend on $NAD^+$ to poly-ADP ribosylate substrates (Krishnakumar & Kraus, 2010), and consequently, HRD cells may require a permanently high $NAD^+$/NADH ratio that is controlled by the metabolic state of the cell. We hypothesized that limiting accessibility to $NAD^+$ could severely

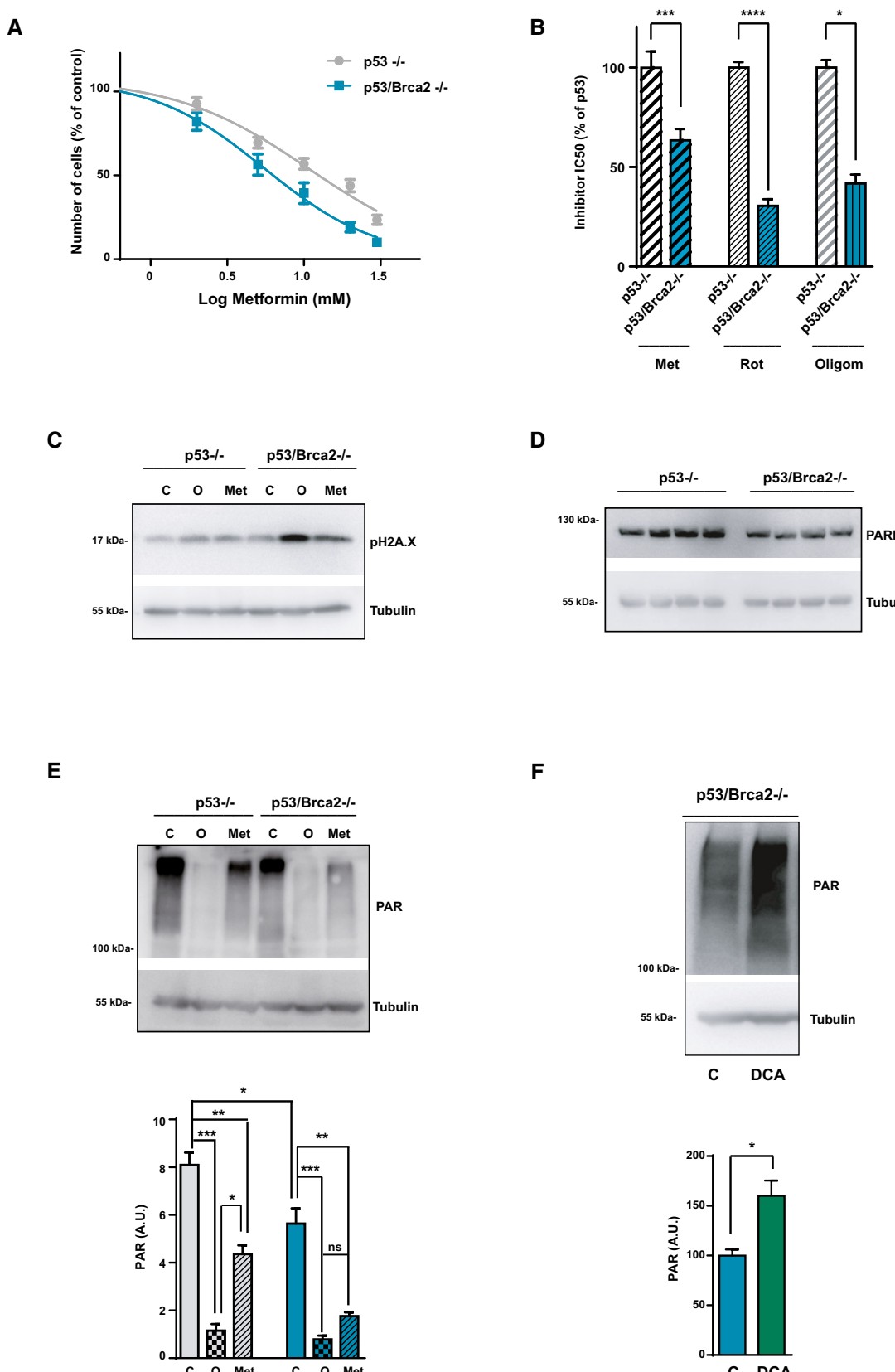

Figure 3.

**Figure 3. HRD cells are more sensitive to metformin.**

A  ID8 *Trp53*-deleted and *Trp53/Brca2*-deleted ovarian tumor cells were incubated for 3 days over a range of metformin concentrations. Cell viability was measured as the frequency of cells revealed by violet crystal staining. Results are expressed relative to the control in the absence of metformin. Each data point represents the mean and SEM of eight independent determinations.

B  ID8 *Trp53*-deleted and *Trp53/Brca2*-deleted ovarian tumor cells were incubated for 3 days over a range of metformin (Met), rotenone (Rot), or oligomycin (Oligom) concentrations. Cell viability was measured as the frequency of cells revealed by violet crystal staining. The $IC_{50}$ was determined relative to ID8 *Trp53*-deleted cells. Each data point represents the mean and SEM of $n = 16$ (Met), $n = 14$ (Rote), or $n = 4$ (Oligom) independent determinations. Statistical significance of two-tailed unpaired Mann–Whitney $U$-tests: *Trp53*-deleted versus *Trp53/Brca2*-deleted in the metformin treatment ***$P = 0.0006$; *Trp53*-deleted versus *Trp53/Brca2*-deleted in the rotenone treatment ****$P = 0.0001$; *Trp53*-deleted versus *Trp53/Brca2*-deleted in the oligomycin treatment *$P = 0.028$.

C  Phospho-histone H2A.X and tubulin expression were measured by immunoblotting in ID8 *Trp53*-deleted and *Trp53/Brca2*-deleted ovarian tumor cells incubated for 48 h with complete DMEM in the absence (C) or presence of 5 mM metformin (Met), or for the last 12 h in the presence of 5 μM olaparib (O). A representative blot of three independent experiments is shown.

D  ID8 *Trp53*-deleted ($n = 4$) and *Trp53/Brca2*-deleted ($n = 4$) ovarian tumor cells grown in complete medium were lysed and immunoblotted using the indicated antibodies. A representative blot of three independent experiments is shown.

E  Top panel: Poly-ADP-ribosylated (PAR) proteins and tubulin (as a loading control) expression were measured by immunoblotting in ID8 *Trp53*-deleted and *Trp53/Brca2*-deleted ovarian tumor cells incubated for 12 h with complete DMEM in the absence (C) or presence of 5 μM olaparib (O) or 5 mM metformin (Met). A representative blot of four independent experiments is shown. Bottom panel: Bars show the relative immunoreactivity of PAR normalized with respect to tubulin in samples from the top. The mean of five independent experiments is shown. Error bars indicate the SEM. Statistical significance of one-way ANOVA, Tukey's multiple comparisons test: *$P < 0.05$; **$P < 0.01$; ***$P < 0.005$.

F  Top panel: Poly-ADP-ribosylated (PAR) proteins and tubulin (as a loading control) expression were measured by immunoblotting in ID8 *Trp53/Brca2*-deleted ovarian tumor cells incubated for 48 h with complete DMEM in the absence (C) or presence of 10 mM dichloroacetate (DCA). A representative blot of five independent experiments is shown. Bottom panel: Bars show the relative immunoreactivity of PAR normalized with respect to tubulin in samples from the top. The mean of five independent experiments is shown. Error bars indicate the SEM. Statistical significance of two-tailed unpaired Mann–Whitney $U$-tests: *$P = 0.0159$.

Source data are available online for this figure.

affect the survival capacity of HRD cells. To test this, the $NAD^+/NADH$ ratio was measured in the cell models after 48 h in the presence or absence of metformin. Relative to *Trp53*-deleted cells, *Trp53/Brca2*-deleted cells showed a 14% greater $NAD^+/NADH$ ratio, mainly due to 46% higher $NAD^+$ levels (Figs 4C and EV5A). Metformin incubation reduced the ratio only in *Trp53/Brca2*-deleted cells (48%, Fig 4C), a drop caused by a decrease in $NAD^+$ and an increase in NADH (Fig EV5A and B).

These results suggest that reduced ATP and $NAD^+$ levels can substantially impair the survival of defective HR cancer cells. To evaluate this hypothesis, we examined the effect of metformin under a range of cell culture conditions. Restriction of carbon sources by incubation with DMEM in the absence of pyruvate increased the sensitivity to metformin in all cancer cell lines tested (Fig 4D), as previously described (Gui *et al*, 2016; Hodeib *et al*, 2018). In fact, differences in metformin's $IC_{50}$ between *Trp53/Brca2*-deleted and *Trp53*-deleted cells disappeared in the absence of pyruvate. However, adding pyruvate to the medium, which raised $NAD^+$ (Gui *et al*, 2016), conferred greater resistance to metformin in both *Trp53*-deleted and *Trp53/Brca2*-deleted cells (Fig 4D) and correlated with higher levels of PAR proteins (Fig 4E). Effects of pyruvate on metformin sensitivity were consistent with changes in ATP levels (Fig EV5C) and in $NAD^+/NADH$ ratio (Fig 4F) associated with pyruvate and/or metformin addition.

The use of $NAD^+$ by poly-ADP-ribosylating enzymes reduces $NAD^+$ levels and releases nicotinamide (NAM). $NAD^+$ levels could be restored by *de novo* synthesis (from tryptophan in *de novo* synthesis pathway) or by salvage pathways from NAM or nicotinic acid (NA) (Canto *et al*, 2015). In addition, changes in cell metabolism also modify available levels of $NAD^+$. To determine the importance of each pathway in the differential response to metformin, we assessed the effect of tryptophan and inhibitors of the enzymes involved in the salvage pathways required for restoring $NAD^+$ levels: GMX1777 (an inhibitor of NAMPT, the key enzyme of NAM

utilization) and 2-hydroxypyridine-3-carboxylic acid (an inhibitor of NAPRT, the key enzyme of NA utilization). Only GMX1777 reduced survival directly in both subsets of ID8 cell (*Trp53/Brca2*-deleted and *Trp53*-deleted) with similar sensitivity ($IC_{50}$ values of 33 nM and 30 nM, respectively). However, when co-administered with metformin, GMX1777 had a differential effect on *Trp53/Brca2*-deleted cells, reducing metformin $IC_{50}$ by 63% compared with a 19% reduction in *Trp53*-deleted cells (Fig 4G). Conversely, when we treated *Trp53/Brca2*-deleted cells with 2 mM nicotinamide, metformin $IC_{50}$ increased (Fig 4H). Finally, analysis of differential cancer cell responses to the NMPRT inhibitor FK866 revealed significantly higher sensibility in cell lines characterized by a high microsatellite instability (MSI-H) phenotype relative to MSI stable and low (Fig 4I). Taken together, these results confirmed that OXPHOS regulates $NAD^+$ levels and the $NAD^+/NADH$ ratio, and that these are key elements of the metformin site of action.

### Metformin blocks growth of HR-defective tumors *in vivo*

This inhibitory capacity of metformin in defective HR cells in culture prompted us to evaluate this inhibitor in the context of ovarian cancer growth *in vivo*. To this end, we first generated palpable tumors from ID8 *Trp53*-deleted or *Trp53/Brca2*-deleted cells injected into the ovary of C57/BL-6 mice. Tumor-bearing mice were then randomized into two groups for one additional month and treated with saline or metformin. Metformin treatment significantly reduced tumor growth (0.12 $cm^3$ versus 0.24 $cm^3$ tumor volume, in treated and control groups, respectively) only in the ID8 *Trp53/Brca2*-deleted cells (Fig 5A and B), as compared to ID8 *Trp53*-deleted cells that were unaffected by metformin treatment (0.13 $cm^3$ versus 0.11 $cm^3$ tumor volume, in control and metformin-treated groups, respectively).

Finally, we evaluated the effects of metformin on orthotopic preclinical models of post-surgery tumor samples from one high-grade

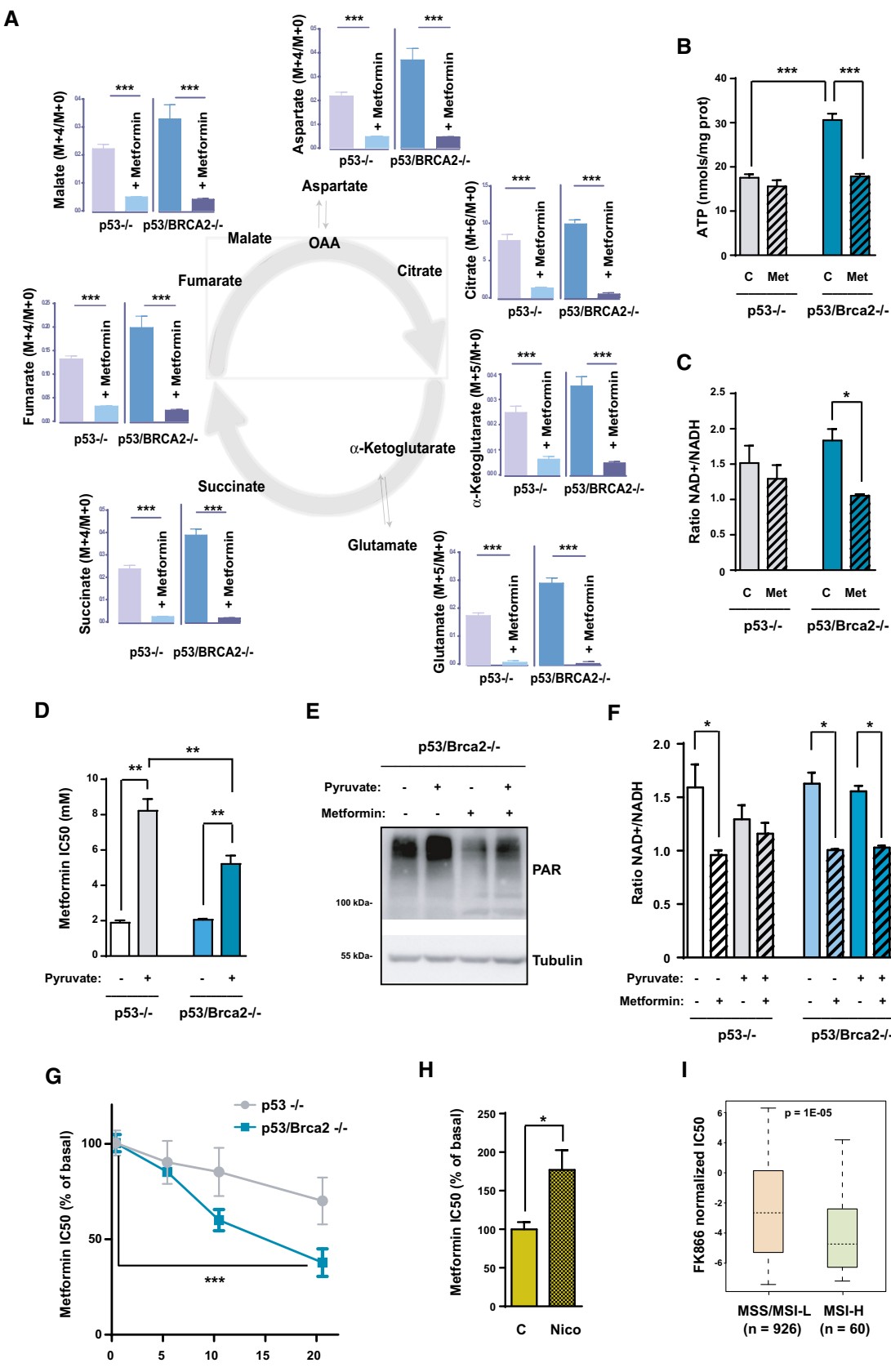

**Figure 4.**

**Figure 4. Metformin affects metabolites of the TCA cycle, ATP, and the NAD$^+$/NADH ratio.**

A Fluxes of glucose carbon into metabolites of the TCA cycle and its branching pathways. The ratio of fully labeled and unlabeled isotopolog amount is shown for each metabolite as analyzed by targeted metabolomics. The mean of four replicates for each cell line is shown. Error bars indicate the SEM. Statistical significance of one-way ANOVA: ***$P < 0.001$.

B ID8 *Trp53*-deleted and *Trp53/Brca2*-deleted ovarian tumor cells were incubated for 48 h with complete DMEM in the absence or presence of 5 mM metformin. Cells were then lysed, and total ATP levels are measured as indicated. Data were normalized with respect to protein content. The mean of seven independent experiments is shown. Error bars indicate the SEM. Statistical significance of two-tailed unpaired Mann–Whitney $U$-tests: ID8 *Trp53*-deleted versus double *Trp53/Brca2*-deleted, ***$P = 0.0006$; ID8 double *Trp53/Brca2*-deleted control versus metformin, ***$P = 0.0006$.

C ID8 *Trp53*-deleted and *Trp53/Brca2*-deleted ovarian tumor cells were incubated for 48 h with complete DMEM in the absence or presence of 5 mM metformin. Cells were then lysed and total NAD$^+$ and total NADH levels measured as indicated. Data were normalized with respect to protein content. Ratios between NAD$^+$ and NADH levels in each situation were calculated. The mean of three (ID8 *Trp53*-deleted, 1.45, 1.97, and 1.13 in the absence and 1.63, 0.96, and 1.29 in the presence of metformin) and four independent experiments (ID8 *Trp53/Brca2*-deleted 1.93, 1.84, 2.17, and 1.40 in the absence and 1.06, 1.10, 0.99, and 1.06 in the presence of metformin) is shown. Error bars indicate the SEM. Statistical significance of two-tailed unpaired Mann–Whitney $U$-tests: ID8 double *Trp53/Brca2*-deleted control versus metformin, *$P = 0.0286$.

D ID8 *Trp53*-deleted and *Trp53/Brca2*-deleted ovarian tumor cells were incubated for 3 days over a range of metformin concentrations in complete DMEM without pyruvate and complete DMEM and 1 mM pyruvate (+Pyr). Cell viability was measured as the frequency of cells stained with crystal violet. The IC$_{50}$ was determined by representing results relative to the control without metformin. Bars represent the mean and SEM of six independent determinations. Statistical significance of one-way ANOVA, Tukey's multiple comparisons test, **$P < 0.01$.

E ID8 *Trp53*-deleted and *Trp53/Brca2*-deleted ovarian tumor cells were incubated for 48 h with complete DMEM without pyruvate or complete DMEM and 1 mM pyruvate in the absence or presence of 5 mM metformin. Cells were lysed and immunoblotted using the indicated antibodies. A representative blot of three independent experiments is shown.

F ID8 *Trp53*-deleted ($n = 3$) and *Trp53/Brca2*-deleted ($n = 3$) ovarian tumor cells were incubated for 2 days in normal DMEM in the absence or presence of 5 mM metformin and in the absence or presence of 1 mM pyruvate. Cells were then lysed and total NAD$^+$ and total NADH levels measured as indicated. Data were normalized with respect to protein content. Ratios between NAD$^+$ and NADH levels in each situation were calculated. Error bars indicate the SEM. Statistical significance of one-way ANOVA, Tukey's multiple comparisons test, *$P < 0.05$.

G ID8 *Trp53*-deleted and *Trp53/Brca2*-deleted ovarian tumor cells were incubated for 3 days over a range of metformin concentrations in complete DMEM, or in complete DMEM and the indicated concentrations of GMX1777 (nM). Cell viability was measured as the frequency of cells revealed by crystal violet staining. The IC$_{50}$ was determined by representing results relative to a zero level of inhibitor. Bars represent the mean and SEM of four independent determinations. Statistical significance of two-tailed unpaired Mann–Whitney $U$-tests: ***$P = 0.0006$.

H ID8 *Trp53*-deleted and *Trp53/Brca2*-deleted ovarian tumor cells were incubated for 3 days over a range of metformin concentrations in complete medium without pyruvate, or in complete medium and 1 mM nicotinamide (+Nico). Cell viability was measured as the frequency of cells stained with crystal violet. The IC$_{50}$ was determined by representing results relative to the control. Bars represent the mean and SEM of six independent determinations. Statistical significance of two-tailed unpaired Mann–Whitney $U$-tests, *$P = 0.028$.

I Box plot showing higher sensibility (i.e., lower standardized half-maximal inhibitory concentration [IC50]) to the NMPRT inhibitor FK866 in cancer cell lines classified as MSI-H (high genomic instability, $n = 60$) relative to MSI stable (MSS) or MSI-low (MSI-L; low genomic instability, $n = 926$). The box plots show a typical display consisting of a median value depicted by the line in the center of the box; an interquartile range (IQR; 25th to the 75th percentile) depicted by the box; the maximum (Q3 + 1.5*IQR) and minimum (Q1 − 1.5*IQR) values depicted by the whisker; and outliers shown as individual circles. The $P$ value of the $t$-test is shown.

Source data are available online for this figure.

---

serous ovarian tumor WT for *BRCA* genes (OVA260 tumor), and one high-grade serous ovarian tumor with a deletion of the exon 20 c.(5243_5277+2788del; 5277+2916_5277+2946delinsGG) of the *BRCA1* gene, implanted in nude mice. Mice bearing these tumors were randomized after implantation into two groups, and when a palpable intra-abdominal mass was detected (3 months), animals were treated with saline or metformin for one additional month. Again, in these PDX models metformin treatment only significantly reduced tumor growth in the *BRCA1* mutated model (Fig 5C and D), with a tumor volume after treatment of 0.37 cm$^3$ in control versus 0.19 cm$^3$ in metformin-treated animals, whereas WT *BRCAs* tumors had a volume post-treatment of 0.66 cm$^3$ versus 0.56 mm$^3$ in control and metformin-treated mice, respectively.

In all, these results confirmed that the effect of metformin on defective HR tumors was also observed *in vivo* and could have consequential implications for the treatment of this kind of cancers.

### Blocking OXPHOS and increasing glycolysis limit the effect of PARP inhibitors

Having established that HRD cells are more sensitive to OXPHOS inhibition, we examined the effect of combining OXPHOS inhibitors with PARP activity inhibitors, such as olaparib. Surprisingly, incubation of *Trp53/Brca2*-deleted ID8 cells with metformin not only had no synergistic activity when combined with olaparib, but it was in fact antagonistic; metformin incubation increased the IC$_{50}$ of olaparib in a dose-dependent manner (Fig 6A), from 0.72 μM in the absence of metformin, to 2.67 and 4.71 μM with 2 and 5 mM metformin, respectively. These results were also observed in *Trp53*-deleted cells (3.29, 3.91, and 5.78 μM, respectively, Fig 6A), leading to equivalent olaparib sensitivity in *Trp53/Brca2*-deleted and *Trp53*-deleted ID8 cells in the presence of 5 mM metformin.

To clarify whether resistance to olaparib was related to the metabolic implications of metformin, we used rotenone, another inhibitor of the respiratory chain. Our results indicated that rotenone also increased the IC$_{50}$ for olaparib (Fig 6B), although it was less effective than metformin. We then examined the effect of metformin on the capacity of olaparib to block PARP activity in *Trp53/Brca2*-deleted cells. As illustrated in Fig 6C, metformin partially blocked the impact of olaparib on the levels of PAR proteins. One of the key effects of PARP inhibitors is to trap PARP proteins at zones with DNA damage (Murai *et al*, 2012). To determine whether metformin could affect this mechanism of action of PARP inhibitors, we evaluated the presence of PARP1 protein in chromatin in the presence or absence of olaparib, or in combination with metformin. Metformin decreased trapping of PARP1 to DNA induced by olaparib (line 3 in the different conditions, Fig 6D).

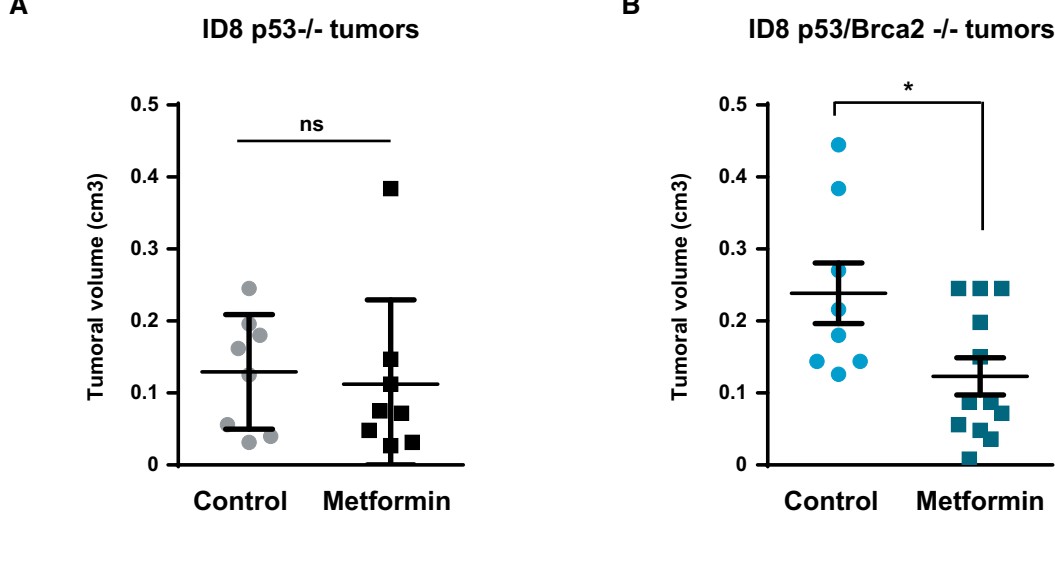

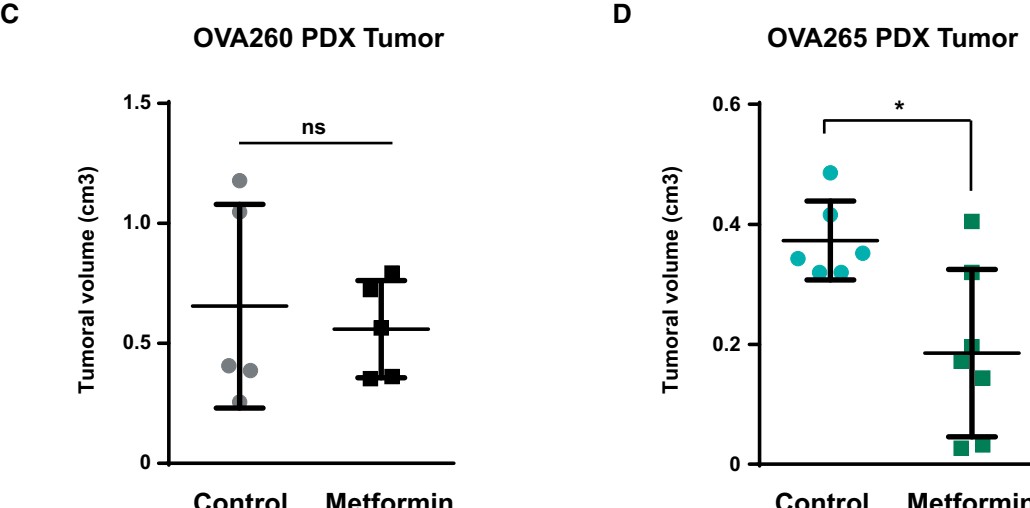

**Figure 5. Metformin blocks growth of HR-defective tumors in vivo.**

A, B   ID8 *Trp53*-deleted (A) and *Trp53/Brca2*-deleted (B) ovarian tumor cells were injected in the ovary of C57/BL-6 mice. After 1 month, animals were treated with vehicle or metformin (100 mg/kg) for 4 weeks. Results are the mean and SEM of 8 controls and 8 metformin-treated animals (A) or of eight controls and 12 metformin-treated animals (B). Statistical significance of two-tailed unpaired Mann–Whitney *U*-tests: *P = 0.0446.

C   Nude mice with orthotopically implanted *BRCA1* WT OVA260 ovarian tumors were treated with vehicle or metformin (100 mg/kg) for 4 weeks. Results are the mean and SEM of five control tumors and five metformin-treated tumors. Statistical significance of two-tailed unpaired Mann–Whitney *U*-test: P = 0.8016.

D   Mice with orthotopically implanted *BRCA1*-mutated OVA265 ovarian tumors were treated with vehicle or metformin (100 mg/kg) for 4 weeks. Results are the mean and SEM of seven control tumors and seven metformin-treated tumors. Statistical significance of two-tailed unpaired Mann–Whitney *U*-tests: *P = 0.0105.

To study which metabolic changes induced by metformin affect olaparib sensitivity, once again, we changed cell culture conditions. Increased olaparib $IC_{50}$ by metformin was mimicked by pyruvate alone, an effect that was more evident in *Trp53/Brca2*-deleted cells (Fig 7A). This confirmed that olaparib resistance was mediated by metabolic perturbations. Metformin in the presence of pyruvate showed the greatest increase in olaparib $IC_{50}$, reaching an equivalent sensitivity to *Trp53*-deleted cells. These results suggested that

PARP inhibitors require active OXPHOS metabolism to influence cell viability. To test this hypothesis, we first measured oxygen consumption in cells incubated for 2 days under the same conditions in which we had analyzed olaparib sensitivity. Metformin more than halved the oxygen consumption rate in all cells and under all conditions analyzed (Fig 7B). In contrast, metformin increased glycolysis in *Trp53*-deleted cells independently of pyruvate, although this increase only occurred in *Trp53/Brca2*-deleted

cells in the presence of pyruvate (Fig 7C). The combination of metformin with pyruvate yielded similar levels of glycolytic flux in *Trp53/Brca2*-deleted and in *Trp53*-deleted cells (Fig 7C). These results suggested that a shift to a more glycolytic metabolism was responsible for olaparib resistance. To confirm this, we decreased glycolysis by incubation with dichloroacetate (DCA). Addition of DCA to *Trp53*-deleted and *Trp53/Brca2*-deleted cells was sufficient to increase sensitivity to olaparib (Fig 7D).

## Tumors that have a glycolytic metabolic profile have a weaker response to olaparib

Our results suggested an alternative mechanism for resistance to olaparib inhibitors in tumors in which OXPHOS is reduced and the glycolytic metabolism in cancer cells is increased. To investigate this possibility, we analyzed expression data from PDX breast cancer models originating from *BRCA1* mutation carriers whose sensitivity

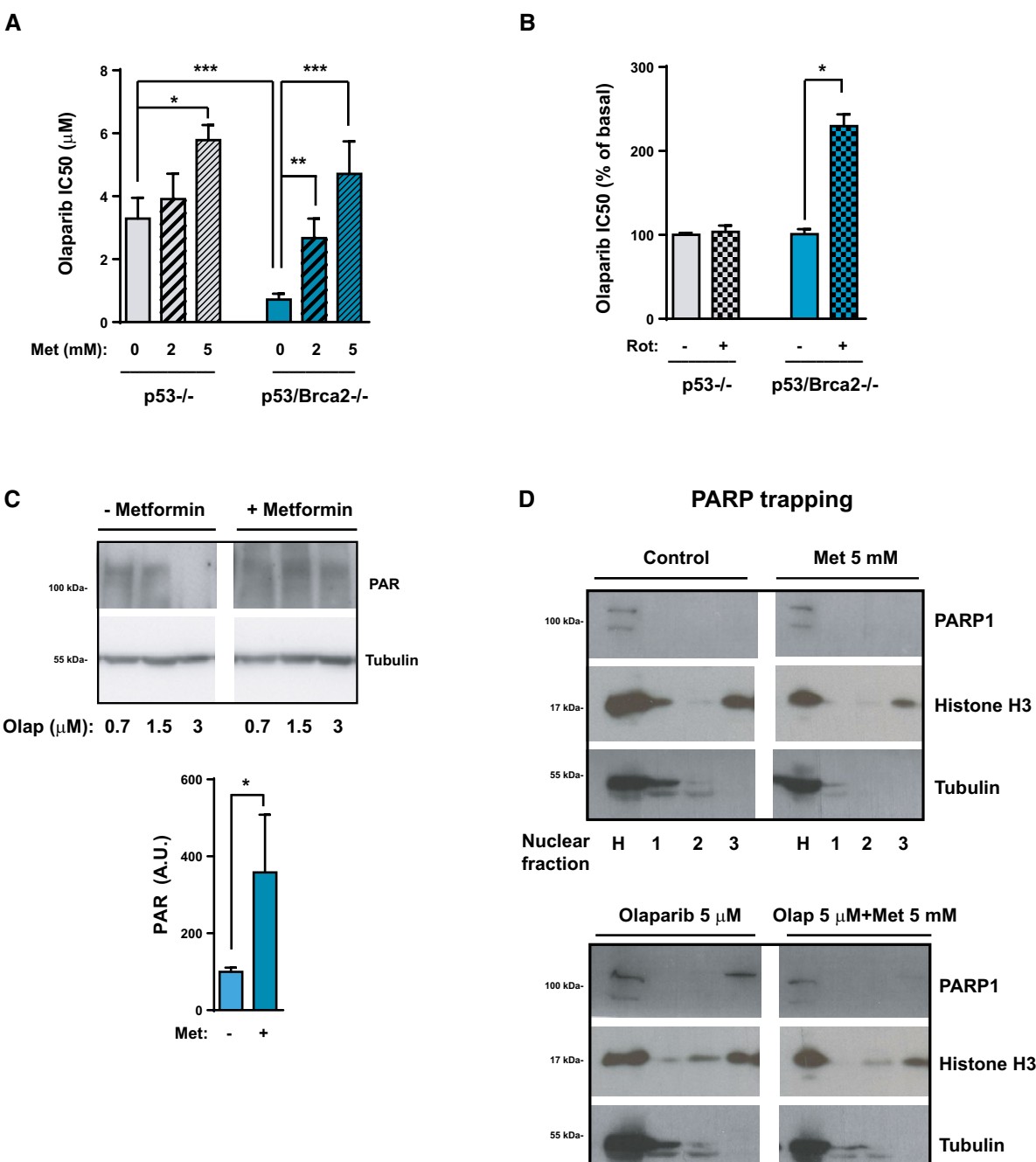

**Figure 6.**

**Figure 6. Metformin reduces olaparib sensitivity.**

A ID8 *Trp53*-deleted and *Trp53/Brca2*-deleted ovarian tumor cells were incubated for 3 days over a range of olaparib concentrations in the absence (0 Met) or presence of 2 mM or 5 mM metformin. Cell viability was measured as the frequency of cells stained with crystal violet. The IC$_{50}$ was determined by representing results relative to the control. Each data point represents the mean and SEM of five independent determinations. Statistical significance of one-way ANOVA ($P < 0.0001$) and two-tailed unpaired Mann–Whitney *U*-tests: ID8 *Trp53*-deleted in the absence of metformin compared with 5 mM metformin *$P = 0.0176$; ID8 *Trp53*-deleted and *Trp53/Brca2*-deleted in the absence of metformin ***$P = 0.0003$; ID8 *Trp53/Brca2*-deleted in the absence of metformin compared with 2 mM metformin (**$P = 0.0023$) or with 5 mM metformin (***$P = 0.0003$).

B ID8 *Trp53*-deleted and *Trp53/Brca2*-deleted ovarian tumor cells were incubated for 3 days over a range of olaparib concentrations in the absence or presence of 5 nM rotenone. Cell viability was measured by counting the number of cells stained with crystal violet. The IC$_{50}$ was determined by representing results relative to the control. Each data point represents the mean and SEM of four independent determinations. Statistical significance of two-tailed unpaired Mann–Whitney *U*-tests: *$P = 0.0286$.

C ID8 *Trp53/Brca2*-deleted ovarian tumor cells were incubated for 48 h with complete DMEM in the presence of 0.7, 1.5, or 3 μM olaparib and the presence or absence of 5 mM metformin. Cells were lysed and immunoblotted using the indicated antibodies. Top panel: A representative blot of four independent experiments is shown. Bottom panel: Bars show the relative immunoreactivity of PAR normalized with respect to tubulin in samples from the top panel (3 μM olaparib). The mean of seven independent experiments is shown. Error bars indicate the SEM. Statistical significance of two-tailed unpaired Mann–Whitney *U*-tests: *$P = 0.0169$.

D ID8 *Trp53/Brca2*-deleted ovarian tumor cells were incubated for 24 h with complete DMEM in the presence or absence of 5 μM olaparib and the presence or absence of 5 mM metformin. Cells were lysed and various nuclear fractions obtained as described. Samples were immunoblotted using the indicated antibodies. A representative blot of three independent experiments is shown.

Source data are available online for this figure.

to olaparib had also been characterized ((Cruz *et al*, 2018) and Appendix Fig S8). In the collection of *BRCA1*-mutant PDXs, a higher level of expression of OXPHOS gene sets and the respiratory chain complex I gene set was detected in the olaparib-sensitive models than in the olaparib-resistant PDX models (Fig 7E). To validate these findings, the levels of expression of the monocarboxylate transporter MCT4 were analyzed by immunohistochemistry. MCT4 expression was lower in these tumors compared with the olaparib-resistant PDXs (Fig 7F). These results confirmed that increased glycolytic metabolism is an alternative mechanism of resistance to PARP inhibition and, therefore, that efficacy of olaparib may be conditioned by the metabolic profile of the cell.

## Discussion

This study demonstrates that cancer cells with HR deficiency undergo a shift from a classical Warburg glycolytic phenotype to one exhibiting increased oxidative metabolism. Normal untransformed cells exposed to acute or chronic DNA damage show increased OXPHOS due to the stimulation of mitochondrial fatty acid oxidation (Brace *et al*, 2016). Given this, it is not surprising that cells that are constantly subjected to genomic instability and problems in the repair systems have adapted their metabolism by increasing oxygen consumption. However, the dependence of HRD cancer cells revealed here highlights a novel cancer vulnerability and provides a complementary explanation of the differential PARP inhibitor responses. Moreover, our results provide a mechanistic explanation, and expand upon previous observations, of the increase in the abundance of TCA intermediates in BRCA1- and p53-deficient cancer models (Breitkopf *et al*, 2017) and of the higher mitochondrial metabolic rate observed in MCF10A *BRCA1* haplo-insufficient cells (Cuyas *et al*, 2016). In the context of HR deficiency, OXPHOS activity proves to be fundamental for maintaining the ATP and NAD$^+$ levels required for the alternative DNA repair mechanisms that ensure cancer survival. The depicted mechanism may also play a role in the emergence of resistance to various DNA-damaging drugs or treatments. Indeed, some oncogenes redirect their metabolism to increase mitochondrial oxidation in order to boost the capacity to detoxify ROS caused by different drugs (e.g., MITF in melanomas) (Vazquez *et al*, 2013), and recent studies have shown increases in OXPHOS metabolism to be associated with resistance to different drugs in various cancer types (Bosc *et al*, 2017; Farge *et al*, 2017; Lee *et al*, 2017). This increase in OXPHOS in HRD cancer cells is caused by a greater capacity to oxidize metabolites, especially glucose through increased PDH activity, without apparent changes in the total mitochondrial mass. In the case of oncogenic transformation, OXPHOS metabolism increases due to the rise in mitochondrial biogenesis that stems from the greater activity of transcription factors such as PGC1α (Vazquez *et al*, 2013) and MYC (Lee *et al*, 2017). Although we did not detect significant changes in PGC1α levels or in mitochondrial mass in any of our models, we cannot rule out the possibility that, in other cancer settings, genomic instability might promote mitochondrial biogenesis.

Why is this increase in OXPHOS metabolism necessary in HRD tumoral cells? Our results suggest that the shift in the metabolic capacity of cells is directed toward maintaining high levels of ATP and NAD$^+$, for the purpose of ensuring PARP-dependent DNA repair activity and thereby cell survival. Cancer cells have a very low NAD$^+$/NADH ratio, and changes in this ratio are more closely associated with tumor growth than is the ATP/ADP ratio (Vander Heiden & DeBerardinis, 2017). In fact, between one-third and 80% of the NAD$^+$ is consumed by PARP enzymes in cells exposed to DNA damage, and the extent of this NAD$^+$ consumption is greater in cells with DNA repair dysfunction than in normally repairing cells (Liu *et al*, 2018). Moreover, normal cells undergo a reduction in NAD$^+$ levels and an increase in oxidative phosphorylation when exposed to DNA damage (Brace *et al*, 2016). If NAD$^+$ levels are reduced, for example, by limiting the carbon sources (glucose or pyruvate) or by inhibiting NAD$^+$ synthesis/recycling, then cancer cells become more sensitive to other stresses, such as cytotoxic agents and PARP inhibitors (Bajrami *et al*, 2012; Chan *et al*, 2014; Piacente *et al*, 2017), or to the blockage of OXPHOS (our results, see later). Conversely, if the NAD$^+$/NADH ratio rises, for example, by incubation with pyruvate or the addition of an exogenous source of NAD$^+$, cells become more resistant to stresses ((Gui *et al*, 2016; Hodeib *et al*, 2018) and our results). This effect and a drop in ATP levels are much more important in cells that depend entirely on PARP to repair DNA damage, as our results show.

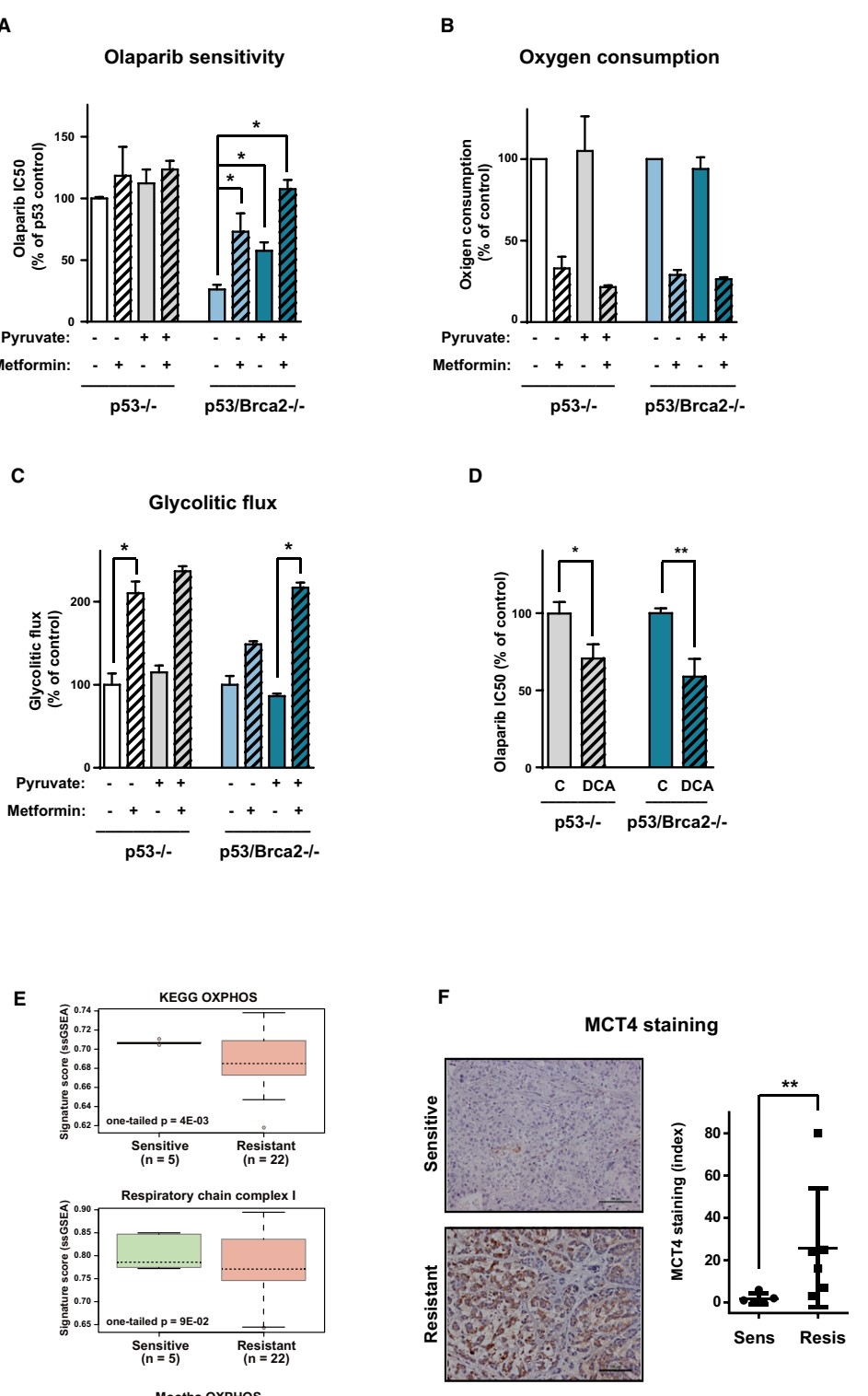

**Figure 7.**

◀

**Figure 7. Increase in glycolytic metabolism is associated with an olaparib-resistant phenotype.**

A ID8 *Trp53*-deleted and *Trp53/Brca2*-deleted ovarian tumor cells were incubated for 3 days in complete DMEM over a range of olaparib concentrations in the absence or presence of 2 mM metformin, and in the absence or presence of 1 mM pyruvate. Cell viability was measured as the frequency of cells stained with crystal violet. The $IC_{50}$ was determined by representing results relative to the control. Each data point represents the mean and SEM of five independent determinations. Statistical significance of two-tailed unpaired Mann–Whitney $U$-tests: *Trp53/Brca2*-deleted in the absence of pyruvate and metformin compared with cells in the absence of pyruvate and the presence of metformin (*$P = 0.0286$), in the presence of pyruvate and the absence of metformin (*$P = 0.0238$), and in the presence of pyruvate and metformin (*$P = 0.0286$).

B Oxygen consumption rates measured using an Oxygraph-2K in ID8 *Trp53*-deleted ($n = 3$) and *Trp53/Brca2*-deleted ($n = 3$) ovarian tumor cells. Cells were incubated for 2 days in normal DMEM in the absence or presence of 2 mM metformin and in the absence or presence of 1 mM pyruvate. Each data point represents the mean and SEM.

C Glycolytic flux was measured in ID8 *Trp53*-deleted ($n = 4$) and *Trp53/Brca2*-deleted ($n = 4$) ovarian tumor cells grown for 2 days in normal DMEM in the absence or presence of 2 mM metformin and in the absence or presence of 1 mM pyruvate. Cells were incubated for 1 h in the presence of $^3$H glucose. Values (in pmol $^3$H $H_2O$ produced) were normalized with respect to protein content. Error bars indicate the SEM. Statistical significance of two-tailed unpaired Mann–Whitney $U$-tests: *Trp53*-deleted cells in the absence of pyruvate and metformin compared with cells in the absence of pyruvate and presence of metformin (*$P = 0.0286$), and in *Trp53/Brca2*-deleted cells comparing cells in the presence of pyruvate and the absence of metformin and in the presence of pyruvate and metformin (*$P = 0.0286$).

D ID8 *Trp53*-deleted and *Trp53/Brca2*-deleted ovarian tumor cells were incubated for 3 days in normal DMEM over a range of olaparib concentrations in the absence or presence of 10 mM dichloroacetate. Cell viability was measured as the frequency of cells stained with crystal violet. The $IC_{50}$ was determined by representing results relative to the control. Each data point represents the mean and SEM of six independent determinations. Statistical significance of two-tailed unpaired Mann–Whitney $U$-tests: in ID8 *Trp53*-deleted (*$P = 0.0303$) and in *Trp53/Brca2*-deleted ovarian tumor cells (**$P = 0.022$).

E Differential signature expression (based on ssGSEA scores) of olaparib-sensitive ($n = 5$) and olaparib-resistant ($n = 22$) PDXs. The results are shown for KEGG OXPHOS, respiratory chain complex I, and Mootha OXPHOS gene sets. The box plots show a typical display consisting of a median value depicted by the line in the center of the box; an interquartile range (IQR; 25th to the 75th percentile) depicted by the box; and the maximum (Q3 + 1.5*IQR) and minimum (Q1 − 1.5*IQR) values depicted by the whisker. Statistical significance of one-tailed unpaired Mann–Whitney $U$-tests is indicated.

F Left panel, MCT4 staining in *BRCA1*-mutated olaparib-sensitive or olaparib-resistant breast cancer PDXs. Scale bars, 100 μm. Right panel, graphs depicting differences in MCT4 staining (multiplicative index) between PDX subgroups (olaparib-sensitive, $n = 5$; olaparib-resistant, $n = 6$). The bars indicate the mean and standard error (SEM). Statistical significance of two-tailed unpaired Mann–Whitney $U$-test, **$P = 0.0087$.

Our work also highlights the importance of OXPHOS inhibitors, such as metformin, in treating *BRCA1/2*-mutated cancers and other types of tumors affected by HR defects. Metformin causes a combination of effects at different levels, such as ROS accumulation, alterations of the redox state, and mitochondrial metabolism, all of which are effects that depend on substrate availability (Liu *et al*, 2018). Our results suggest that metformin alters these pathways in *BRCA1/2*-mutated cells, but those that explain the differential effect on HRD cells are the alteration of the redox state and of the ATP levels, leading to a drop in the capacity to repair damaged DNA. In other cancer cells, with wild-type status of *BRCA1/2*, metformin causes an increase in NADH levels, but not as much in $NAD^+$ (Liu *et al*, 2016; Hodeib *et al*, 2018). In our study, due to the extensive consumption of $NAD^+$ by PARP enzymes in HRD cells, we observed a combined effect, a drop in $NAD^+$ levels after short-term metformin treatment and an increase in NADH over the long term. The beneficial effects of biguanidines (metformin, phenformin, etc.) in the treatment of some cancers have been evaluated epidemiologically (Evans *et al*, 2005; Jiralerspong *et al*, 2009; Libby *et al*, 2009; Decensi *et al*, 2010) and at the molecular level (Liu *et al*, 2016; Bosc *et al*, 2017). In fact, other OXPHOS inhibitors are being developed to treat cancer (Bosc *et al*, 2017; Molina *et al*, 2018). Our study depicts a mechanistic model for their clinical rationale in the context of HR deficiency.

A second highlight of our work is that a high rate of glycolytic metabolism decreases the effect of olaparib on tumoral cells due to a reduction in the capacity of olaparib to block PARP activity and poly-ADP protein ribosylation. Our results indicate that by modifying the metabolic capacity, either by inhibiting complex I with metformin or rotenone, or by adding or not adding pyruvate or DCA, the sensitivity of cells to olaparib is modified. The molecular mechanism involved seems to be related to alterations in the capacity of olaparib to block PARP activity and trap PARP on the chromatin, one of the major mechanisms by which PARP inhibitors

work (Murai *et al*, 2012; Shen *et al*, 2015). Another explanation could be that the alteration of cell cycle caused by metformin affects olaparib's capacity to block PARP. In fact, DNA repair protein expression depends on progression during the cell cycle. WEE1 tyrosine kinase blocks cell cycle progression, and its inhibition increases olaparib sensitivity (Garcia *et al*, 2017). Metformin, by blocking cell cycle and impairing alternative DNA repair mechanisms to HR, should cause the opposite effect, i.e., decreasing olaparib sensitivity. Moreover, a high glycolytic rate has been shown to protect cells from apoptosis induced by deprivation of growth factors (Vander Heiden *et al*, 2001; El Mjiyad *et al*, 2011). By increasing glycolysis, metformin could alter the capacity of olaparib to induce apoptosis.

As a consequence of the poor effect of PARP inhibitors in cells that shifted to a more glycolytic metabolism, increasing glycolysis emerges as an alternative mechanism for reducing olaparib sensitivity. Several mechanisms involved in innate or generated resistance to PARP inhibitors have been described (reviewed in Ref. (Bitler *et al*, 2017; Liao *et al*, 2018)), affecting inhibitor efflux (Rottenberg *et al*, 2008), restoration of PARP function (Gogola *et al*, 2018) or homologous repair mechanisms (Norquist *et al*, 2011; Barber *et al*, 2013), overstimulation of other DNA repair mechanisms (Panier & Boulton, 2014), and alteration of DNA replication (Ray Chaudhuri *et al*, 2016). Our results imply that the metabolic profile of a tumor also determines its sensitivity to olaparib. If a tumor presents a high rate of glycolytic metabolism it will be relatively insensitive to olaparib, while oxidative metabolism assures a strong response to the drug. Thus, analysis of the metabolic profile of a tumor will give a good indication of olaparib response and highlight a new way of selecting patients in whom PARP inhibitors are less effective. Our results also indicate that blocking glycolysis resensitizes tumors to olaparib, providing the basis for possible alternatives to avoid resistance in these situations.

# Materials and Methods

### Chemical compounds

Olaparib was obtained from LC Laboratories (Woburn, MA) and was diluted in DMSO. Cell culture media, FBS, glutamine, pyruvate, and antibiotics were obtained from Gibco (Thermo Fisher Scientific, Waltham, MA). Metformin (cat. D150959), nicotinamide (Cat. 72340), GMX1777, rotenone (Cat. R8875), dichloroacetate (Cat. 347795), and other reagents were of analytical or molecular biology grade and were purchased from Sigma-Aldrich (St Louis, MO).

### Patients

Samples and data from patients in this study were provided by the IDIBGI Biobank (Biobanc IDIBGI, B.0000872) and from the tumor tissue bank of the Hospital of Bellvitge-IDIBELL. They were processed following standard operating procedures, with the appropriate approval of the Ethics and Scientific Committees and conformed to the principles set out in the WMA Declaration of Helsinki. A signed informed consent was obtained from each patient.

### ID8Trp53-deleted or Trp53/Brca2-deleted cells *in vivo* studies

ID8 *Trp53*-deleted or *Trp53*/*Brca2*-deleted cells were used to generate orthotopic ovarian tumors in 7-week-old female C57/BL-6 mice. Mice were anesthetized with isoflurane, and 2 millions of cells were injected into the ovary in 20 μl of DMEM. When a palpable intra-abdominal mass was detected (1 month), animals were randomized into two groups with similar tumor volumes and metformin (100 mg/kg) was administered intraperitoneally every day from Monday to Friday for 4 weeks. Control group mice received intraperitoneal sterile serum under the same schedule as metformin-treated mice. All mice also received a subcutaneous injection of 500 μl of 5% glucose solution. The treatments had no significant effect on mouse body weight, and the animals appeared healthy and active throughout the study. Tumor response was evaluated at the moment of sacrifice by determining tumor volume and calculated as (length) (width$^2$/2). All the animal studies were approved by the local committee for animal care (IDIBELL, DAAM 5766).

### High-grade ovarian serous PDX

Female nu/nu Swiss mice (strain NMRI-*Foxn1*$^{nu}$/*Foxn1*$^{nu}$) were purchased from Janvier (Saint Berthevin, France). Mice were housed and maintained in laminar flow cabinets under specific pathogen-free conditions. All the animal studies were approved by the local committee for animal care (IDIBELL, DAAM 5766).

The ovarian OVA200 and OVA265 used were perpetuated in 6-week-old female nude mice by consecutive passages. Clinically (FIGO classification), the primary tumor of both OVAs was classified as stage III high-grade ovarian serous tumors. OVA265 presented a deletion of the exon 20 c.(5243_5277+2788del; 5277+2916_5277+2946delinsGG) in the *BRCA1* gene. No *BRCA1 or BRCA2* mutations were detected in the OVA260 tumor.

When a palpable intra-abdominal mass was detected (3 months), animals were randomized into two groups with similar tumor volumes and metformin (100 mg/kg) was intraperitoneally administered every day from Monday to Friday for 4 weeks. Control group mice received intraperitoneal sterile serum under the same schedule as metformin-treated mice. All mice also received a subcutaneous injection of 500 μl of 5% glucose solution. The treatments had no significant effect on mouse body weight, and the animals appeared healthy and active throughout the study. Tumor response was evaluated at the moment of sacrifice by determining tumor volume and calculated as (length) (width$^2$/2).

### Breast PDX

Breast BRCA1 patient-derived tumor xenografts (PDXs) have been described elsewhere (Cruz *et al*, 2018).

### Immunohistochemistry in paraffin-embedded samples and scoring

Paraffin-embedded sections were deparaffinized in xylene and rehydrated in downgraded alcohols and distilled water. Antigens were retrieved under high-pressure conditions for 4 min in citrate buffer, pH 6. In the case of NDUFV2 detection, a pepsin digestion was also performed. Endogen peroxidases were deactivated, and samples were blocked with goat normal serum before primary antibody incubation overnight at 4°C. 1/70 rabbit polyclonal anti-NDUFV2 antibody (Cat. NBP1-84475, Novus Biologicals, Littleton, CO) and 1/100 rabbit polyclonal anti-MCT4 (SLC16A3) (Cat. NBP1-81251, Novus Biologicals, Littleton, CO) were used. Sections were incubated with the specific secondary anti-rabbit EnVision antibodies (Dako, Agilent Technologies, Santa Clara, CA), followed by the DAB developing system (Dako). Samples were counterstained with hematoxylin and visualized under light microscopy.

MCT4 and NDUFV2 were measured on the following grading scale: no detectable signal (0 points); low-intensity signal (1 point); moderate-intensity signal (2 points); and high-intensity signal (3 points). The labeling frequency was scored as the percentage of positive tumor cells. The multiplicative index of intensity and labeling frequency was used in our analysis, as previously described (Alsina-Sanchis *et al*, 2016).

### Cell culture and media

The ID8 murine ovarian cancer cells used have been described previously (Walton *et al*, 2016). We compared two genetically engineered cell lines: ID8 *Trp53*-deleted and *Trp53*/*Brca2*-deleted ID8 cells. Cells were cultured in Dulbecco's modified Eagle's medium (DMEM, 25 mM glucose, 1 mM pyruvate, Sigma-Aldrich) supplemented with 4% fetal bovine serum (GIBCO), L-glutamine, ITS (5 μg/ml insulin, 5 μg/ml transferrin, and 5 ng/ml sodium selenite), and antibiotics (100 U/ml penicillin and 0.1 mg/ml streptomycin). In some experiments, we used DMEM complete medium without pyruvate and supplemented or not with 1 mM pyruvate. SKOV-3 (bought to Sigma-Aldrich in 2014, passages 5–20) ovarian cancer human cells, murine ovarian cancer cells obtained from tumors generated in mice with combinations of genetic alterations (*Trp53*$^{-/-}$, *myc*, and *Akt* or *Trp53*$^{-/-}$, *Brca1*$^{-/-}$, *myc*, and *Akt*) (Xing & Orsulic, 2006) and murine *Trp53*-deficient or double *Trp53*/*Tsc2*-deficient MEFs were cultured in DMEM (Sigma-Aldrich),

supplemented with 10% fetal bovine serum (GIBCO), L-glutamine, and antibiotics (100 U/ml penicillin and 0.1 mg/ml streptomycin). All cells were grown at 37°C in a humidified atmosphere with 5% $CO_2$, and mycoplasma was tested by PCR every month.

To generate deletion of the *BRCA2* gene in SKOV-3, we used GenCRISPR™ gRNA/Cas9 lentiCRISPR v2 vector (GenScript, Piscataway, NJ) targeting the *BRCA2* sequence CTGTCTACCTGAC CAATCGA. SKOV-3 cells were infected with this vector or an empty vector. After 2 days, cells were treated with puromycin (5 µg/ml) and clones were isolated (SKOV-3-SC for the empty vector and SKOV-3-BRCA2$^{-/-}$ 1.2). Validation of the *BRCA2* gene deletion was performed by sequencing (STAB VIDA, Caparica, Portugal), increase in the olaparib sensitivity (2.8 µM in SKOV-3-BRCA2$^{-/-}$ 1.2 compared with 6.9 µM SKOV-3-SC), increase in the basal DNA damage (staining for phospho-histone H2A.X Ser139), and loss of the ability to form RAD51 foci after olaparib treatment compared with the control cells (Appendix Fig S4 and (Walton *et al*, 2016)).

### Cell viability assay

Cells were plated in 96-well plates, 750 cells per well, in eightplicate. Cells were treated with the various compounds and respective solvents for 3 days. Cell viability was determined as the final number of cells detected on the cell culture plate, once they had been fixed with methanol and crystal violet had been incorporated (0.01%). Results were confirmed by measuring metabolic activity as indicated by the methyl-thiazol-tetrazolium (MTT) assay (Sigma, St Louis, MO). Briefly, when the treatment was finished, 10 µM MTT was added to each well and incubated for an additional 4 h. The blue MTT formazan precipitate was dissolved in DMSO, and the optical density was measured (absorbance at 570 nm) on a multi-well plate reader.

### Mitochondrial respiration assay (Oxygraph)

Oxygen consumption was measured at 37°C in a high-resolution oxygraph (Oxygraph-2k; Oroboros Instruments, Innsbruck, Austria). Cells were seeded and grown to 60–70% confluence. They were then detached with accutase and resuspended in medium at a final concentration of $1 \times 10^6$ cells/ml. Basal oxygen consumption was measured in the closed respirometer chamber and expressed as the value per million viable cells. Oxygen flux, which is directly proportional to oxygen consumption, was recorded continuously using DatLab software 4.3 (Oroboros Instruments, Innsbruck, Austria).

### Lactate production and glucose consumption assays

Cells were seeded in 12-well plate ($3 \times 10^5$ cells per well) in quadruplicate. After 24 h, the medium was collected for analysis of L-lactate and glucose. Cells were collected to check the protein content for normalization. The concentration of L-lactate was determined using an enzymatic reaction based on the oxidation of L-lactate to pyruvate, as previously described (Alvarez *et al*, 2016). Lactate production was calculated as the difference between lactate concentrations at 0 and 24 h. The concentration of glucose was determined with the PGO (glucose oxidase/peroxidase) enzyme system (cat. P7119, Sigma-Aldrich) and o-dianisidine as a colorimetric substrate. The assay was performed in accordance with the manufacturer's

instructions. Glucose consumption was calculated as the difference between glucose concentration at 0 and 24 h.

### Glycolytic rate measurement

The glycolytic rate was determined by measuring tritium release from D-[5-$^3$H] glucose to [$^3$H] water in the reaction catalyzed by enolase. Briefly, cells were seeded in 12-well plates ($3 \times 10^5$ cells per well) and, before measuring, incubated in basal medium with 1 µCi of D-[5-$^3$H] glucose (American Radiolabeled Chemicals, St. Louis, MO) per milliliter of medium for 60 min. After incubation, 50 µl aliquots of medium were transferred to uncapped microtubes containing 50 µl of 0.2 N HCl. Each tube was placed in a scintillation vial containing 0.5 ml of cold $H_2O$, avoiding contact between the water in the vial and the contents of the microtube. The vials were sealed to allow the [$^3$H] water to evaporate from the microtube and condense in the 0.5 ml of water at the bottom of the scintillation vial. Diffusion took place for 24 h at 37°C. The amount of diffused and undiffused [$^3$H] water was determined by scintillation counting. A control sample containing no cells was used to correct the [$^3$H] water in the D-[5-$^3$H]glucose. In addition, a sample containing a known quantity of [$^3$H] water was used to determine the completeness of diffusion. Protein content was used for normalization.

### NAD$^+$ and NADH measurement

The concentrations of NAD$^+$ and NADH were determined with the NAD/NADH-Glo™ Assay (cat. G9071, Promega, Madison, WI), following the manufacturer's instructions. Protein content was used for normalization.

### ATP levels

ATP concentration was determined using the StayBrite™ Highly Stable Luciferase/Luciferin Reagent (cat. K790, BioVision, Milpitas, CA), following the manufacturer's instructions. Protein content was used for normalization.

### Metabolomics

ID8 *Trp53*-deleted and *Trp53/Brca2*-deleted cells were treated with or without 5 mM metformin for 48 h in complete medium. The last 12-h medium was changed to basal DMEM, with or without metformin and without pyruvate and glutamine, in the presence of 25 mM D-Glucose (U-13C6, Cat. CLM-1396-1, Cambridge Isotope Laboratories Inc, Andover, MA). Afterward, cells were rinsed with cold PBS, detached, centrifuged at 300 *g* for 5 min, and the pellet frozen in liquid nitrogen.

Metabolites were extracted by adding 300 µl of cold methanol/water (8:1, v:v). Samples were vortexed for 30 s and immersed in liquid $N_2$ to disrupt cell membranes followed by 10 s of bath sonication. These two steps were repeated three times. Cell lysates were incubated for 20 min in ice before centrifugation (5,000 *g*, 15 min at 4°C). Ten microliter of 13C-glycerol (150 ppm) was added to the supernatant as internal standard. Next, 250 µl of each sample was dried under a stream of $N_2$ gas. Lyophilized polar extracts were incubated with 50 µl methoxyamine in pyridine (40 µg/µl) for

45 min at 60°C. To increase volatility of the compounds, we silylated the samples using 25 μl N-methyl-N-trimethylsilyltrifluoroacetamide with 1% trimethylchlorosilane (Thermo Fisher Scientific) for 30 min at 60°C. A 7890 A GC system coupled to a 7000 QqQ mass spectrometer (Agilent Technologies, Palo Alto, CA.) was used for isotopologue determination. Derivatized samples were injected (1 μl) in the gas chromatograph system with a split inlet equipped with a J&W Scientific DB5−MS+DG stationary phase column (30 mm × 0.25 mm i.d., 0.1-μm film, Agilent Technologies). Helium was used as a carrier gas at a flow rate of 1 ml/min in constant flow mode. The injector split ratio was adjusted to 1:5, and oven temperature was programmed at 70°C for 1 min and increased at 10°C/min to 325°C. The ionization performed was positive chemical ionization (CI) with isobutene as reagent gas. Mass spectral data on the 7000 QqQ were acquired in scan mode monitoring selected ion clusters of the different metabolites.

## Mitochondrial mass detection assay (MitoTracker)

Cells were seeded in six-well plates at a density of 50,000 cells per well. When cells reached the desired confluency (50–80%), culture medium was removed, and pre-warmed (37°C) staining solution containing the non-potential-dependent MitoTracker™ Green FM dye (1 μM) was added. Plates were incubated for 30 min at 37°C. MitoTracker staining was analyzed by flow cytometry. Experiments were carried out in triplicate for each set.

## Mitochondrial (MitoSOX) superoxide production assay

Cells were seeded in six-well plates at a density of 50,000 cells per well and grown to 50–80% confluence. MitoSOX Red solution was added to a final concentration of 5 μM, in accordance with the manufacturer's recommendation. Cells were allowed to load MitoSOX Red for 30 min and then washed three times with Hank's buffered salt solution (HBSS) containing calcium and magnesium. For flow cytometry analysis, after 15 min of MitoSOX loading, all three cell lines were trypsinized for 4 min and neutralized with medium. Cells were centrifuged for 5 min at 300 g, resuspended with HBSS containing 1% BSA/EDTA, and aliquoted into sterile FACS tubes. MitoSOX Red was excited at 510 nm. Flow cytometry was performed at different times. Experiments were carried out in duplicate for each set.

## Flow cytometry assay of phospho-histone H2A.X (Ser 139)

Cells were seeded in six-well plates at a density of 50,000 cells per well and grown to 50–80% confluence. All cell lines were trypsinized for 4 min and then washed three times with PBS. Cells were centrifuged for 5 min at 600 g, resuspended in PFA 4%, and aliquoted 100 μl into sterile FACS tubes. After 15-min incubation at RT in the dark, cells were centrifuged and washed with PBS. Cells were resuspended in 100 μl of Saponin buffer (3× Saponin buffer: PBS, 1% BSA, 0.1% sodium azide, pH 7.4), and 5 μl of goat serum was added during 15 min as a blocking step (final concentration of 5%). Cells were centrifuged and washed using 1–2 mL of 1× Saponin buffer solution and centrifuged 2 min at 600 g. One hundred microliter of 1× Saponin buffer solution was added with 1/100 rabbit polyclonal anti-phospho-histone H2A.X (Ser139)

antibody (Cat. ab11174, Abcam, Cambridge, UK) and incubated 30 min at RT in the dark. Cells were centrifuged and washed using 1–2 ml of 1× Saponin buffer solution, and resuspended in 100 μl of 1× Saponin buffer solution and the secondary Alexa Fluor 488 goat anti-rabbit (Invitrogen, Thermo Fisher). After 30-min incubation at RT in the dark, cells were centrifuged and washed using 1–2 ml of 1× Saponin buffer solution, resuspended in 100 μl of 1× Saponin buffer solution, and staining was analyzed by flow cytometry. Experiments were carried out in duplicate for each set.

## Apoptosis analysis by flow cytometry

Cells ($2 \times 10^5$) were seeded in 100-mm culture plates and, after 24 h, were treated with 5 μM olaparib or 5 mM metformin for 2 days. Afterward, cells were detached, centrifuged at 300 g for 5 min, and resuspended at a density of $5 \times 10^5$ cells/ml in 1× PBS-1% FBS. Then, 100 μl of the cell suspension was mixed with Annexin-V kit buffer (1:1; Muse™ Annexin-V & Dead Cell Assay, Merck Millipore, Merck KGaA, Darmstadt, Germany). After 20 min of incubation at room temperature, cells were examined in a Muse™ Cell Analyzer (Merck Millipore). All conditions were assessed in two independent experiments.

## Cell cycle

After 2 days cell of culture with or without 2 mM metformin, cells were detached, counted, and 1 million cells were centrifuged at 600 g for 5 min. The cell pellet was resuspended in 100 μl citrate buffer (250 mM sucrose, 40 mM trisodium citrate dihydrate, pH 7.6) and treated with acotase. Samples were stained with propidium iodide on ice for 10 min. Cells were analyzed in a FACSCalibur Flow Cytometer running ModFit software.

## Western blotting

Cells were lysed using RIPA lysis buffer and Western blot was performed, as described elsewhere (Alsina-Sanchis et al, 2016). Blots were incubated with 1/500 rabbit polyclonal anti-NDUFV2 antibody (Cat. NBP1-84475, Novus Biologicals, Littleton, CO), 1/500 rabbit polyclonal anti-MCT4 (SLC16A3) (Cat. NBP1-81251, Novus Biologicals, Littleton, CO), 1/1,000 rabbit polyclonal anti-phospho-histone H2A.X (Ser139) antibody (Cat. 2577, Cell Signaling Technologies), 1/500 mouse monoclonal anti-VDAC1 antibody (Msa03, MitoSciences, Abcam, Cambridge, UK), 1/1,000 rabbit polyclonal anti-GLUT-1 (Cat. Ab652, Abcam, Cambridge, UK), 1/500 rabbit polyclonal anti-TIGAR antibody (Cat. Santa Cruz, CA), 1/500 rabbit polyclonal anti-phospho-PDH-E1a (Ser 293) antibody (Cat. AP1062, Calbiochem, Millipore), 1/1,000 rabbit polyclonal anti-PDH-E1a antibody (Cat. AP1062, Calbiochem, Millipore), 1/1,000 rabbit polyclonal anti-phospho-ACC (Ser79) antibody (Cat. 3661, Cell Signaling Technologies), 1/1,000 rabbit polyclonal anti-ACC antibody (Cat. 3676, Cell Signaling Technologies), 1/1,000 rabbit polyclonal anti-PARP1 antibody (Cell Signaling Technologies), 1/1,000 monoclonal mouse anti-poly-ADP-ribose (PAR) antibody (Cat. AM80, Calbiochem, Millipore), 1/1,000 rabbit polyclonal anti-histone H3 antibody (Cat. Ab1791, Abcam), 1/2,500 monoclonal mouse anti-α-tubulin antibody (Sigma-Aldrich), or 1/2,500 monoclonal mouse anti-vinculin antibody (Sigma-Aldrich) in TBS containing 1% non-fat dry milk overnight at 4°C. Volumetric analysis was

carried out using the Quantity One volume analysis tool (Bio-Rad, Hercules, CA).

### Nuclear extracts fractions

We adapted a protocol based on Murai *et al* (2012). Six millions of cells were scrapped in cold PBS, centrifuged at 300 *g* for 5 min, and resuspended in 150 µl hypotonic buffer (10 mM HEPES, pH 7.4, 1 mM EDTA, and 0.5 mM $MgCl_2$) with protease inhibitors (0.1 µg/ml benzamidine, 100 µM phenylmethylsulfonyl fluoride, 1 µM pepstatin A, 1 µg/ml leupeptin, 4 µg/ml aprotinin). After 5 min at 4°C, 150 µl hypotonic buffer containing 30% sucrose was added and then centrifuged at 12,000 *g* at 4°C 10 min. The pellet was resuspended in 150 µl buffer A (50 mM HEPES, pH 7.4, 100 mM KCl, 2.5 mM $MgCl_2$, 0.05% Triton X-100 and protease inhibitors). This was H fraction (total nuclear fraction). Seventy-five microliter of H was centrifuged at 12,000 *g* at 4°C 10 min. The supernatant was fraction 1. The pellet was resuspended in 75 µl buffer B (50 mM HEPES, pH 7.4, 250 mM KCl, 2.5 mM $MgCl_2$, 0.05% Triton X-100 and protease inhibitors) and centrifuged at 12,000 *g* at 4°C 10 min. The supernatant was fraction 2. The pellet was resuspended in 75 µl buffer C (50 mM HEPES, pH 7.4, 500 mM KCl, 2.5 mM $MgCl_2$, 0.1% Triton X-100 and protease inhibitors) and centrifuged at 12,000 *g* at 4°C 10 min. The supernatant was fraction 3.

### Quantitative real-time PCR

Total RNA from cells was extracted using TRIzol Reagent (Ambion, Thermo Fisher Scientific). cDNA was obtained from a reverse transcription reaction (High-Capacity cDNA Reverse Transcription Kit; Applied Biosystems, Life Technologies, CA). Real-time PCR of cDNA obtained was carried out on a LightCycler instrument (Roche Molecular Biochemicals, Lewes, UK). After initial incubation at 95°C for 10 min, 40 cycles of amplification were performed with denaturation at 95°C for 10 s, followed by annealing for 20 s (at 64°C) and extension at 72°C for 13 s. The ΔCt values were calculated after subtracting the mean Ct values of the β-actin gene from *LdhB* or *Hk2* mean Ct values. Results are presented as values of 2(−ΔΔCt) relative to ID8 *Trp53*-deleted cells values.

The mouse-specific primers used were *LdhB* (5′-CATTGCGTCCG TTGCAGATG and 5′-GGAGGAACAAGCTCCCGTG), *Hk*2 (5′-TGATC GCCTGCTTATTCACGG and 5′-AACCGCCTAGAAATCTCCAGA), and the housekeeping gene β-actin (5′-GGGGGTTGAGGTGTTGAG and 5′-GTCTCAAGTCAGTGTACAGGCC).

### Gene expression analysis

Pre-processed and normalized data from breast and high-grade serous ovarian tumors were taken from the NCI Genomic Data Commons data portal. Somatic mutation data were obtained following approval by the Data Access Committee (project #11689). The results published here are partly based upon data generated by The Cancer Genome Atlas (TCGA) (Cancer Genome Atlas Research Network, 2011), managed by the National Cancer Institute (NCI, Bethesda) and National Human Genome Research Institute (NHGRI, Bethesda). Information about TCGA can be found at http://cancergenome.nih.gov. Mutational signatures were defined using the R mutSignatures package. The gene set enrichment tool, GSEA (Subramanian *et al*, 2005), was used with

default values selected for all parameters, and employing curated KEGG gene sets, and the *t*-statistic of the gene expression differences as a ranking metric. The expression signature scores were computed using the ssGSEA algorithm with standard parameters and considering all genes included in the OXPHOS gene set. ssGSEA gene expression correlations were computed in R. Normalized gene expression data from ovarian tumors treated with carboplatin in the CTCR-OV01 study (GEO reference GSE15622) were analyzed using unpaired *t*-tests; these assessments provided the statistic ranks used in subsequent GSEAs with standard parameters.

The BRCA1/2 mutation status in primary breast and ovarian tumors from TCGA studies was determined by annotations downloaded from the cBioPortal combined with curated information from two publications (Kraya *et al*, 2019; Yost *et al*, 2019).

The non-TCGA ovarian cancer dataset included 152 patients with high-grade serous ovarian cancer cared at the Cedars-Sinai Medical Center, Ovarian Cancer Research Alliance, New York, and compiled information included age at diagnosis, tumor stage, progression-free survival, and overall survival. Total RNA from formalin-fixed paraffin-embedded (FFPE) samples of omental metastases from these 152 chemotherapy naïve patients was isolated using the miRNeasy FFPE kit according to the manufacturer's instruction (Qiagen). RNA expression of genes included in the PanCancer Progression Panel, Immunology Panel, and custom-selected genes was analyzed by NanoString nCounter technology. Data were normalized with nSolver software. To infer the KEGG OXPHOS signature score in these tumors, we used as surrogate genes whose expression is highly correlated with this signature in RNAseq data from TCGA ovarian cancer. Ten genes were selected that showed PCCs > 0.55 with KEGG OXPHOS in TCGA data; then, these genes were used as a ssGSEA signature to infer OXPHOS level in the independent ovarian cancer dataset. Multivariate Cox regression analyses using the survfit and survminer packages in R software were applied.

Pre-processed and normalized basal gene expression and $IC_{50}$ responses in cancer cell lines were taken from the Genomics of Drug Sensitivity in Cancer dataset (Iorio *et al*, 2016). The expression signature scores were computed using the ssGSEA algorithm calculated in the GSVA package in Bioconductor (Hanzelmann *et al*, 2013).

### Statistical analyses

Statistical analyses were carried out with SPSS for Windows 13.0 (SPSS, Inc., Chicago, IL) or GraphPad Prism 6.01 and in the R statistical environment. Differences between experimental and control groups were determined using the Mann–Whitney *U*-test, Student's *t*-test, or the ordinary one-way ANOVA analysis. Animals in which tumor implantation did not succeed (around 10–20% in every implantation process) were excluded from the analysis and not included in the final figure. Animals were grouped by similar tumor size after palpation and assigned to control (animals treated with vehicle) or metformin treatment randomly. No blinding was performed.

## Data and software availability

Metabolomics: We include two Excel files as source data with the metabolomic data we obtained (Figs 2F and 4A).

## The paper explained

### Problem

Metabolism and the generation of reactive oxygen species (ROS) contribute to the acquisition of DNA mutations and genomic instability in cancer. But how genomic instability influences the metabolic capacity of cancer cells is nevertheless poorly understood.

### Results

Here, we show that homologous recombination-defective (HRD) and in general genomic instable cancers rely on oxidative metabolism to supply key metabolites important for PARP-dependent DNA repair mechanisms. Studies in breast and ovarian cancer HRD models depict a metabolic shift that includes repression of the glycolytic phenotype and enhanced expression of the oxidative phosphorylation (OXPHOS) pathway and its key components. In consequence, HRD cells are more sensitive to inhibitors of OXPHOS, as metformin. On the other hand, shifting from an OXPHOS to a highly glycolytic metabolism interferes with the sensitivity to PARP inhibitors in these HRD cells, emerging as a new mechanism that determines PARP inhibitor sensitivity.

### Impact

This study shows a mechanistic link between two major cancer hallmarks, which in turn suggests novel possibilities for specifically treating HRD cancers with OXPHOS inhibitors.

The RNAseq datasets produced in this study are available at GEO (Gene Expression Omnibus, NIH) under the accession number GSE148310 (https://www.ncbi.nlm.nih.gov/geo/query/acc.cgi?acc=GSE148310).

**Expanded View** for this article is available online.

## Acknowledgements

This study has been funded by the Ministerio de Ciencia, Innovación y Universidades, which is part of the Agencia Estatal de Investigación (AEI), through the project SAF2017-85869-R (cofunded by the European Regional Development Fund (ERDF), a way to build Europe) to FV and BFU2015-66030-R to JCP; by the FIS PI15/00854 and FIS PI16/01898 (Instituto Carlos III, cofunded by FEDER funds/European Regional Development Fund (ERDF), a way to build Europe) to MAP and AVillanueva and with the support of the Secretariat for Universities and Research of the Department of Business and Knowledge of the Government of Catalonia (2017SGR449) to FV. We thank the CERCA Program/Generalitat de Catalunya for their institutional support. We particularly wish to acknowledge the collaboration of the patients and the IDIBGI Biobank (Biobanc IDIBGI, B.0000872), which is part of the Spanish National Biobank Network and the Xarxa de Bancs de Tumors de Catalunya (XBTC), financed by the Pla Director d'Oncologia de Catalunya, Spain. We thank Cristina Saura (Breast Cancer & Melanoma Group) and people from the Experimental Therapeutics Group at VHIO and Sara González (Unitat de Diagnòstic Molecular, ICO-Duran i Reynals). We thank H. Simon, R. Bartrons, and A. Manzano (Universitat de Barcelona) and A. Vaquero (IDIBELL) for reagents.

## Author contributions

Conception and design: AL, MAP, JCP, FV; Acquisition of data (provided animals, acquired and managed patients, provided facilities, etc.): AL, PH, AF, DG, RM, VS-C, IM, VS, CL, PB, JB, JM, XM-G, AVid, AVil, BT-H, HT, SO, AJ, OY, CM-P, LP, MAP, FV; Analysis and interpretation of data (e.g., statistical analysis, biostatistics, computational analysis): AL, LP, MAP, JCP, FV; Writing, review, and/or revision of the manuscript: AL, CM-P, MAP, JCP, FV; Study supervision: MAP, JCP, FV.

## Conflict of interest

This study was supported by unrestricted grant from Roche to finance the ProCure Programme, which was paid to the Catalan Institute of Oncology (2015).

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
