## [Review Process File · EMBO Molecular Medicine]

Tumors defective in homologous recombination rely on oxidative metabolism: Relevance to treatments with PARP inhibitors

Álvaro Lahiguera, Petra Hyroššová, Agnès Figueras, Diana Garzón, Roger Moreno, Vanessa Soto-Cerrato, Iain McNeish, Violeta Serra, Conxi Lazaro, Pilar Barretina, Joan Brunet, Javier Menéndez, Xavier Matias-Guiu, August Vidal, Alberto Villanueva, Barbie Taylor-Harding, Hisashi Tanaka, Sandra Orsulic, Alexandra Junza, Oscar Yanes, Cristina Muñoz-Pinedo, Luís Palomero, Miquel Àngel Pujana, José Carlos Perales & Francesc Viñals

Review timeline:

Submission date:	26th Jul 2019
Editorial Decision:	16th Sep 2019
Revision received:	27th Feb 2020
Editorial Decision:	17th Mar 2020
Revision received:	5th Apr 2020
Accepted:	9th Apr 2020

Editor: Celine Carret

Transaction Report:

1st Editorial Decision

16th Sep 2019

Thank you for the submission of your manuscript to EMBO Molecular Medicine. We have now heard back from the two referees whom we asked to evaluate your manuscript.

As you will see from the reports below, the referees find the topic of your study of potential interest. However, they both feel that the use of poorly controlled models negatively impact on the study. Because only the ID8 model is informative, we would like to invite you to confirm your findings using another isogenic model as well as validate data in PDXs. Both referees agreed that these should be the minimum requirements for further consideration of the article for publication in EMBO Molecular Medicine. Should you be able to, using metabolomics and isotopic tracing for this study, as recommended by ref.2, would also greatly strengthen the findings. I also would like to mention that while both referees agree that applicability appears limited considering the apparent resistance to PARPi induced by metformin, referee 2 suggest a different interpretation of the data since resistance is expected to be reverted in a setting of fasting (point 5 of referee 2's revision).

We would therefore welcome the submission of a revised version within three months for further consideration and would like to encourage you to address all the criticisms raised as suggested to improve conclusiveness and clarity. Please note that EMBO Molecular Medicine strongly supports a single round of revision and that, as acceptance or rejection of the manuscript will depend on another round of review, your responses should be as complete as possible.

***** Reviewer's comments *****

Referee #1 (Comments on Novelty/Model System for Author):

Only one relevant murine model is used ID8 p53/BRCA2-/- . MCF7 and MDA-MB-231 are not good models for genomic instability comparison. Similarly, unless the mechanism of cisplatin-

resistance in the SKOV-3-R pair is known, this is not a good Olaparib sensitive/responsive model system to use. Overall, the choice of good, well defined, isogenic model systems is a major limitation.

Referee #1 (Remarks for Author):

Summary of work and major conclusions

Authors present work aimed at understanding how OXPPOS is altered and what metabolic dependencies are specific to homologous recombination (HR) defective breast and ovarian cancers. In brief, the approach identified OXPPOS alterations in HR-defective models then used this information to propose targeting HR defective cancers with metformin. Author provide evidence to suggest that metformin decreases PARP1 trapping and OXPPOS is lower in PARP resistant tumours.

Is the question addressed important?

The paper addresses an important question in examining how OXPPOS is altered in HR defective breast and ovarian cancer and whether these vulnerabilities can be potentially exploited to enhance the efficacy of PARP inhibitor treatment currently given to these HR defective patients.

Are the conclusions novel and will they influence thinking in the field?

The observation that metformin decreases PARP1 trapping is novel. However, it is already known that PARP1 is an NAD⁺ dependent enzyme and as such this diminishes the novelty of the observation. Furthermore, the observation that metformin alters PARP1 trapping needs to be validated more robustly and in a number of relevant HR competent / defective models and using orthogonal methods for measuring PARP1 trapping eg micorirradiation. Treatment wise, the paper suggests not to combine metformin and PARP inhibitors and that a decrease in OXPPOS could be a potential resistance mechanism to Olaparib treatment.

Quality of the data provided

In general, most of the observations are limited to one murine isogenic system (ID8) - the other comparisons e.g. MCF7 (genomically stable) vs MDA-MB-231 (genomically unstable) do not make sense - there are a lot of differences between these 2 cell lines that have nothing to do with their ability to repair DNA; same is true for the SKOV platinum sensitive/resistant pair. Overall, the choice, and variety, of isogenic model systems could be improved upon.

Leaps of faith are made in places and additional experiments must be performed to validate robustness of the observations in additional relevant models. Some context also needs to be given regarding the scale of the perturbations seen in the ID8 p53/BRCA2^{-/-} model (for example, the changes in glucose, lactate etc - how do these compare to a known modifier of glucose or lactate consumption in terms of scale?). No BRCA2 rescue experiments have been performed to confirm that the modest difference observed in OXPPOS is due to loss of BRCA2. No in vivo data provided, which is a limitation.

Major points

- Only one relevant murine model is used ID8 p53/BRCA2^{-/-} . MCF7 and MDA-MB-231 are not good models for genomic instability comparison. Similarly, unless the mechanism of cisplatin-resistance in the SKOV-3-R pair is known, this is not a good Olaparib sensitive/responsive model system to use. Overall, the choice of good, well defined, isogenic model systems is a major limitation.
- A BRCA2 rescue experiment in the ID8 p53/BRCA2^{-/-} cell line is needed to confirm that loss of BRCA2 is responsible for the modest OXPPOS differences observed.
- Figure 2A and B - Is the observed increase in O₂ consumption due to proliferation alterations or genomic instability? The paper states that proliferation is not the cause but authors do not present

the cell line proliferation data.

- Provide some sort of control for glucose/lactate/glycolytic assays / provide some context to help explain the significance of the observed decrease in activity in the ID8 p53/BRCA2^{-/-} model
- If the authors are suggesting upregulation of TIGAR is responsible for inhibiting glycolysis in the p53/BRCA2^{-/-} model, knocking down TIGAR and examining a reversion of the metabolic changes observed seems critical
- There are some leaps of faith made - why were MCT4 and NDUFV2 examined in figure 1 ? Complex I is a large protein complex made up of 45/46 different subunits. Why did they look at this subunit in particular, was it enriched in the GSEA analysis? Did they look at other Complex I subunits?
- No substantial in vivo validation of observed biology - the analysis of the PDX data seems limited

Minor points

- Figure 5D - PARP trapping western; cytoplasmic fractionation loading controls needed to rule out contamination in nuclear fraction
- Supplementary Figure 4A - quantification of mitotracker would make comparison easier and scale bars for reference
- Supplementary Figure 4B-D - Vinculin wrote out in full for consistency
- Supplementary Figure 4D - looks like VDAC is more highly expressed in SKOV-3 compared to SKOV3-R and paper states unaffected mitochondria mass is seen in models ?
- Supplementary Figure 5B - GLUT1 appears glycosylated in HR competent models but this is not commented on ?
- Figure 6B - spelling mistake Oxigen -> Oxygen

Referee #2 (Remarks for Author):

The manuscript by Lahiguera et al. identifies OXPHOS as a new vulnerability in tumors characterized by deficient homologous recombination (HR). Mechanistically, the dependency on mitochondrial respiration of HR deficient cells is mediated by the increased demand of NAD⁺ and ATP necessary to sustain PARP activity. Furthermore, shifting metabolism from oxidative to glycolytic would reduce sensitivity to PARP inhibitors. Although this is a very interesting study there are some key points the authors have to address before the manuscript is suitable for publication.

- 1) It is not clear if the phenotype described by the authors is a general feature of genomically unstable tumors or a specific characteristic of HR deficient cells, as title and data suggest. The authors should be consistent throughout the manuscript and avoid to jump from one to another.
- 2) A major concern is about the models used. Besides ID8, that is the most reliable and controlled one since cells harbor specific genomic events in a well-defined genomic background, all the other models are questionable and poorly matched. As an example, how the authors take into consideration genomic differences between MCF7 and MDA-MB-231. I would suggest to focus the manuscript on ID8 model only and move all the other data to supplementary figures. It would be ideal to have another isogenic model, maybe breast, and repeat at least some key experiments.
- 3) Instead of partial assessments of metabolic pathways, an exhaustive representation of the metabolic changes in HR deficient cells would be better captured by metabolomics and isotopic

tracing. Can the authors consider these approaches?

4) Another critical point not addressed by the authors is why HR defective cells are not sensitive to other ETC inhibitors. If continuous supplementation of NAD⁺ and ATP is necessary to sustain PARP activity, as hypothesized by the authors, the same effect would be expected irrespective to which ETC complex is targeted. Did authors attempt to inhibit complex V? Are ROS really not involved in this phenotype? One would expect that increased ROS further damage DNA making HR deficient cells exquisitely sensitive to ETC inhibitors. Because a key point, authors have to repeat experiments in Fig.3 using rotenone, antimycin and oligomycin. Mitochondrial ROS should be quantified by flowcytometry and data represented as histograms and mean of fluorescence for all the ETC inhibitors. Specifically, ROS quenching by NAC should be demonstrated. DNA damage and oxidation should also be assessed by phosphorylation of H2AX (FACS, IF) and 8-oxo-dG staining. Moreover, does GSH depletion through buthionine sulphoximine sensitize cells to OXPHOS inhibitors? PAR should be evaluated in response to rotenone, antimycin and oligomycin as in Fig. 3G.

5) The authors describe that metformin treatment makes cells partially resistant to olaparib, as well as pyruvate. Does metformin exert a similar effect on olaparib IC₅₀ in low glucose? It is possible that the resistance to olaparib can be simply mediated by an increased NAD⁺ regeneration through LDH in response to metformin and pyruvate. Can the authors measure lactate and assess NAD⁺/NADH ratio in this setting? If this hypothesis is correct a synergistic effect of metformin and olaparib should be expected in low glucose, a condition that can be explored in the clinic through fasting.

6) Authors should attempt to validate the findings in vivo using PDXs.

Minor points.

- a) In Figure 1A, why do authors use expression as a surrogate of function Instead of mutational status?
- b) Differences in cell cycle between WT and HR deficient cells can be responsible for the described differences in OXPHOS and glycolysis. Although on page 8 authors claim there are no differences across models, differences are substantial in the workhorse model ID8 where double KO cells appear to be 38% slower than their trp53-deleted counterpart. Although these differences do not appear from Sup. Fig. 8A, the authors should provide cell cycle data and better comment on these results.
- c) Mitochondrial mass has not been quantified appropriately. A non-potential-dependent mitotracker (green) should be used and data quantified by FACS and represented as mean fluorescence. Poor quality immunoblots in Sup. Fig.8 should be replaced and immunofluorescence analysis for VDAC1 in ID8 should be considered.
- d) Figure 2G, show protein level as well.
- e) All the experiments are normalized on protein content? Can authors show data for cell number (DNA) at least for oxygen consumption.
- f) On page 4, the sentence "in this setting, inefficient ATP production due to incomplete oxidation of glucose is compensated by greater glucose consumption" is misleading. As reported in the cited references the Warburg effect is not the results of energetics rather the biosynthetic demand of proliferating cells.
- g) The authors referred to Figure 5D,E instead of 5C,D.

1st Revision - authors' response

27th Feb 2020

Referee #1

Major points

• **Only one relevant murine model is used ID8 p53/BRCA2^{-/-}. MCF7 and MDA-MB-231 are not good models for genomic instability comparison. Similarly, unless the mechanism of cisplatin-resistance in the SKOV-3-R pair is known, this is not a good Olaparib**

sensitive/responsive model system to use. Overall, the choice of good, well defined, isogenic model systems is a major limitation.

We agree with the reviewer that the work quality would improve with a better focus on only relevant models. For this reason, we have eliminated data from MCF7 and MDA-MB-231 from our article and we show mainly isogenic cell models. Thus, we have maintained all the data on ID8 p53 ^{-/-} or p53/Brca2 ^{-/-} cells and MEF p53 ^{-/-} or p53/TSC2 ^{-/-}. Moreover, we have also included new data obtained on two new ovarian cancer cell models:

First, we have generated a new human SKOV-3 derived cell line in which we have deleted the *BRCA2* gene by using a lentiviral GenCRISPR™ gRNA/Cas9 lentiCRISPR v2 vector (GenScript, Piscataway, NJ) (SKOV-3 *BRCA2*^{-/-} 1.2). SKOV-3 cells already present mutated p53. We generated a control SKOV-3-SC for the empty vector and SKOV-3-*BRCA2*^{-/-} 1.2. Validation of the *BRCA2* gene deletion was performed by sequencing, increase in the olaparib sensitivity, increase in the basal DNA damage (staining for phosphoHistone H2A.X Ser139) and loss of the ability to form RAD51 foci after olaparib treatment compared with the control cells (new Sup. Fig. 4).

We have also included new data concerning Brca1 deleted cells, obtained on new ovarian cancer cells obtained in Sandra Orsulic's laboratory (David Geffen School of Medicine at UCLA, Los Angeles, CA (USA)). These cells presented *Trp53*^{-/-}, *myc* and *Akt* or *Trp53*^{-/-}, *Brca1*^{-/-}, *myc* and *Akt* genetic alterations (Xing & Orsulic, Cancer Res. 2006) These cells were isolated from ovarian tumors after intraperitoneal injection of cells in nude mice.

In all these models with alterations in HR or genomic instability we have observed an increase in oxygen consumption and high PDH activation (new Extended View Figures 1 and 2 in the manuscript).

As suggested by the reviewers, we reserved the main figures of our manuscript for the ID8 based cell model and moved all the data from other cell lines to the Supplementary Data.

• A BRCA2 rescue experiment in the ID8 p53/BRCA2^{-/-} cell line is needed to confirm that loss of BRCA2 is responsible for the modest OXPHOS differences observed.

Unfortunately, we failed to recover the expression of Brca2 on ID8 p53/Brca2 ^{-/-} cells. As an alternative experiment, we obtained MDA-MB-436 from a collaborator overexpressing or not Brca1 by retroviral expression. The cell line MDA-MB-436 presents the 5396 + 1G>A mutation in the splice donor site of exon 20, classified as pathogenic mutations, and accompanied by loss of the other BRCA1 allele (Eldstrot et al. Cancer Res. 2006; 66(1): 41-5). Cells overexpressing the WT form of BRCA1 presented less sensitivity to olaparib compared with the normal MDA-MB-436. Oxygen consumption in these cells did not unveiled differences between Brca1 ^{-/-} and Brca1 overexpressing cells.

These results suggest that changes on OXPHOS could be caused, not by direct genetic regulation by Brca proteins or the complex that they form, but by the increase in chronic DNA damage, as suggested by experiments of other authors using normal untransformed cells (Brace et al, 2016, NPJ Aging Mech Dis 2: 16022).

• Figure 2A and B - Is the observed increase in O₂ consumption due to proliferation alterations or genomic instability? The paper states that proliferation is not the cause but authors do not present the cell line proliferation data.

Differences on cell proliferation between ID8 p53^{-/-} and p53/Brca2^{-/-} cells exist, as confirmed by growing the same number of cells in culture plates and counting the number of cells at various times. A 30% lower number of cells was observed in exponentially growing p53/Brca2^{-/-} cells compared with p53^{-/-} cells at 48h and 72h.

The difference in cell number was similar to the changes observed by MTT or by crystal violet staining described in the manuscript and could be associated to the metabolic changes observed. We measured cell proliferative capacity in the different cell types studied and observed various scenarios as indicated in the table:

CELLS	GENETIC MODIFICATION	PROLIFERATION INDEX MODIFICATION	OXYGEN CONSUMPTION MODIFICATION
ID8 Mouse ovarian cancer	Trp53/Brca2 ^{-/-} versus Trp53 ^{-/-}	32% ↓	27% ↑
Mouse ovarian cancer (Orsulic Laboratory)	Trp53/Brca1 ^{-/-} , myc and Akt overexpression versus Trp53 ^{-/-} , myc and Akt overexpression	45% ↓	24% ↑
SKOV-3 resistant human ovarian cancer	Unknown	21% ↑	49% ↑
SKOV-3 BRCA2 deleted human ovarian cancer	TRP53/BRCA2 ^{-/-} versus TRP53 ^{-/-}	75% ↓	47% ↑
MEF mouse fibroblasts	Trp53/Tsc2 ^{-/-} versus Trp53	12% ↓	100% ↑

It is true that all the HRD cell models in culture present low proliferative rates compared with their respective non HRD models, probably caused by alterations in the cell cycle progression that we observed by flow cytometry analysis. Thus, we cannot discard that proliferation alterations could participate in the metabolic changes observed in our cell culture models. It is for this reason that we eliminate the sentence concerning the lack of effect of proliferation in our new version of the manuscript. But is also true that when we tried to confirm this in a more relevant human cancers setting (by Ki67 immunohistochemistry in the same ovarian WT or Brca2 tumoral patient samples analyzed in Fig. 1D and 1E showing significant changes on MCT4 and NDUFV2) we did not observe changes on tumoral proliferation in these patient samples (new Sup. Fig. 1B). Therefore, we discarded changes on cell proliferation as the main reason for the metabolic changes observed in HRD tumors.

Additionally, even considering the possibility of an effect of proliferation on the metabolic adaptations found, it is important to stress that our work clearly show that by modulating metabolism we are able to modify PARP activity, for example using complex I inhibitors (lowering OXPHOS decreases PARP activity and decreases cell viability), with pyruvate or with DCA (increasing OXPHOS is sufficient to increase PARP activity).

• Provide some sort of control for glucose/lactate/glycolytic assays / provide some context to help explain the significance of the observed decrease in activity in the ID8 p53/BRCA2-/- model

The significant changes in glycolytic flux or lactate production observed in p53/Brca2 -/-; 18% decrease in glycolytic flux and a 15% decrease in lactate production are consistent with the amplitude of the global metabolic changes described, probably as a consequence of an adaptation to the high requirements of ATP and NAD⁺ in cells with a high DNA repair activity.

These changes are of similar magnitude to those described in the literature by other metabolic effectors in tumoral cells. For example, the transcription factor BACH1 controls metabolic enzymes and ETC genes expression decreasing glucose utilization in the TCA and OXPHOS metabolism. Decreasing BACH1 expression shifts breast cancer cells to a more OXPHOS dependent metabolism, with quantitative changes that are equivalent to our p53/Brca2 -/- cells. Decrease in BACH1 levels only causes a drop in of 15% in lactate level, for example (Lee et al., Nature 568: 254, 2019, Extended Data Fig. 2B).

• If the authors are suggesting upregulation of TIGAR is responsible for inhibiting glycolysis in the p53/BRCA2-/- model, knocking down TIGAR and examining a reversion of the metabolic changes observed seems critical

In order to explore a possible role of TIGAR in our experimental setting, we modified expression of TIGAR by siRNA treatment (decreasing TIGAR levels) or by stable overexpression (doubling TIGAR levels), but we did not observed changes in OXPHOS in any of these cells. Moreover, when we analyzed TIGAR gene expression and OXPHOS genes expression data from the breast and ovarian cancer studies of the TCGA, we did not find a significant correlation between them. In all, these results suggest to us that we should consider changes in TIGAR expression more as a consequence of a metabolic adaptation, not as the cause of the metabolic changes observed. We have obtained preliminary data suggesting that the changes on OXPHOS proteins and other metabolic enzymes consequence of DNA damage could be induced by transcription factors such as BACH1, a protein that, as mentioned before, controls OXPHOS gene expression.

• There are some leaps of faith made - why were MCT4 and NDUFV2 examined in figure 1 ? Complex I is a large protein complex made up of 45/46 different subunits. Why did they look at this subunit in particular, was it enriched in the GSEA analysis? Did they look at other Complex I subunits?

Unfortunately, we failed to properly explain this dataset in the manuscript, and hence the spot-on criticisms by the reviewer and others. We chose NDUFV2 subunit of the complex I for two reasons:

- First, because this subunit was one of the genes that were increased in the HR defective associated S3+ signature that contribute to the GSEA analysis.
- Second, antibodies against this protein were validated in immunohistochemistry analysis by other authors.

Concerning MCT4, the election was based in its role on lactate extrusion from tumoral cells described in the bibliography and in the availability of high-quality antibodies to perform immunohistochemical analysis.

• No substantial in vivo validation of observed biology - the analysis of the PDX data seems limited

We agree with the reviewer. To tackle the issue we are providing in the revised manuscript with a more suitable and complete set of experiments in vivo by using 2 additional models:

- Intraovarian injection of ID8 p53-/-or p53/Brca2-/- cells was used to grow tumors until they achieved a palpable size, with subsequent treatment with metformin or vehicle. Our results clearly indicate that p53/Brca2-/- tumors are more sensitive to metformin than p53 -/- tumors, confirming the results obtained in cell culture conditions.

- Ovarian orthotopic PDX models, in which we compared a patient derived BRCA1 mutated ovarian tumor PDX with a patient WT BRCA1 ovarian tumor derived PDX. Again, the BRCA1 mutated PDX was more sensitive to metformin compared with the WT tumor, confirming our in vitro results. All these data have been incorporated in the new Figure 5 of the revised manuscript.

Minor points

• Figure 5D - PARP trapping western; cytoplasmic fractionation loading controls needed to rule out contamination in nuclear fraction.

As suggested by the reviewer, in the new version of new Figure 6 we incorporated immunoblot for tubulin as a cytoplasmic marker. As shown in the figure, the amount of tubulin decreases with increased astringency in the wash steps, until finally disappears in fraction 3, where we detected the presence of PARP1.

• **Supplementary Figure 4A - quantification of mitotracker would make comparison easier and scale bars for reference**

The reviewer is right, so we reevaluated this part of the manuscript by measuring mitochondrial mass with a membrane potential-independent mitotracker and quantification by flow cytometry. Data is presented in the new Sup. Fig 6 suggesting a non-significant change on mitochondrial mass between p53/Brca2 *-/-* and p53 *-/-* cells.

• **Supplementary Figure 4B-D - Vinculin wrote out in full for consistency**

The reviewer is right and, as indicated in the previous point, we modified the quantification and interpretation of the data in the revised manuscript.

• **Supplementary Figure 4D - looks like VDAC is more highly expressed in SKOV-3 compared to SKOV3-R and paper states unaffected mitochondria mass is seen in models ?**

The reviewer is again right, but we have no explanation for these changes.

• **Supplementary Figure 5B - GLUT1 appears glycosylated in HR competent models but this is not commented on ?**

The differences in the glycosylation level observed have not been studied in detail, but it is possible that differences in N-acetyl glucosamine provision could explain these differences.

• **Figure 6B - spelling mistake Oxigen -> Oxygen**

We corrected this mistake in the manuscript.

Referee #2 (Remarks for Author):

The manuscript by Lahiguera et al. identifies OXPHOS as a new vulnerability in tumors characterized by deficient homologous recombination (HR). Mechanistically, the dependency on mitochondrial respiration of HR deficient cells is mediated by the increased demand of NAD⁺ and ATP necessary to sustain PARP activity. Furthermore, shifting metabolism from oxidative to glycolytic would reduce sensitivity to PARP inhibitors.

Although this is a very interesting study there are some key points the authors have to address before the manuscript is suitable for publication.

1) It is not clear if the phenotype described by the authors is a general feature of genomically unstable tumors or a specific characteristic of HR deficient cells, as title and data suggest. The authors should be consistent throughout the manuscript and avoid to jump from one to another.

We completely agree with the reviewer on this. Even though we hypothesize that the conclusions of the present work will probably concern non HR-defective, genomic instable cells, our data is limited to HR-defective models, and so our interpretation will focus mainly on those datasets.

2) A major concern is about the models used. Besides ID8, that is the most reliable and controlled one since cells harbor specific genomic events in a well-defined genomic background, all the other models are questionable and poorly matched. As an example, how the authors take into consideration genomic differences between MCF7 and MDA-MB-231. I would suggest to focus the manuscript on ID8 model only and move all the other data to supplementary figures. It would be ideal to have another isogenic model, maybe breast, and repeat at least some key experiments.

We agree with the reviewer that the work quality would improve with a better focus on only relevant models. For this reason, we have eliminated data from MCF7 and MDA-MB-231 from our article and we show mainly isogenic cell models. Thus, we have maintained all the data on ID8 p53 *-/-* or p53/Brca2 *-/-* cells and MEF p53 *-/-* or p53/TSC2 *-/-*. Moreover, we have also included new data obtained on two new ovarian cancer cell models:

First, we have generated a new human SKOV-3 derived cell line in which we have deleted the *BRCA2* gene by using a lentiviral GenCRISPR™ gRNA/Cas9 lentiCRISPR v2 vector (GenScript, Piscataway, NJ) (SKOV-3 *BRCA2*^{-/-} 1.2). SKOV-3 cells already present mutated p53. We generated a control SKOV-3-SC for the empty vector and SKOV-3-*BRCA2*^{-/-} 1.2. Validation of the *BRCA2* gene deletion was performed by sequencing, increase in the olaparib sensitivity, increase in the basal DNA damage (staining for phosphoHistone H2A.X Ser139) and loss of the ability to form RAD51 foci after olaparib treatment compared with the control cells (new Sup. Fig. 4).

We have also included new data concerning *Brca1* deleted cells obtained on new ovarian cancer cells obtained in Sandra Orsulic's laboratory (David Geffen School of Medicine at UCLA, Los Angeles, CA (USA)). These cells presented *Trp53*^{-/-}, *myc* and *Akt* or *Trp53*^{-/-}, *Brca1*^{-/-}, *myc* and *Akt* genetic alterations (Xing & Orsulic, Cancer Res. 2006) These cells were isolated from ovarian tumors after intraperitoneal injection of cells in nude mice.

In all these models with alterations in HR or genomic instability we have observed an increase in oxygen consumption and high PDH activation (new Extended View Figures 1 and 2 in the manuscript).

As suggested by the reviewers, we reserved the main figures of our manuscript for the ID8 based cell model and moved all the data from other cell lines to the Supplementary Data.

3) Instead of partial assessments of metabolic pathways, an exhaustive representation of the metabolic changes in HR deficient cells would be better captured by metabolomics and isotopic tracing. Can the authors consider these approaches?

We thank the reviewer for the comment and suggestion. Following her/his recommendation we have performed a metabolomic study using ¹³C-glucose as a tracer. Results obtained (incorporated in the new Fig. 2F and new Fig. 4A) confirmed our initial observations indicating a higher flux of carbons towards the Krebs cycle in the p53/*Brca2*^{-/-} cells. Moreover, incubation in the presence of metformin causes a blockade in OXPHOS metabolism of both p53/*Brca2*^{-/-} and p53^{-/-} cells.

4) Another critical point not addressed by the authors is why HR defective cells are not sensitive to other ETC inhibitors. If continuous supplementation of NAD⁺ and ATP is necessary to sustain PARP activity, as hypothesized by the authors, the same effect would be expected irrespective to which ETC complex is targeted. Did authors attempt to inhibit complex V? Are ROS really not involved in this phenotype? One would expect that increased ROS further damage DNA making HR deficient cells exquisitely sensitive to ETC inhibitors. Because a key point, authors have to repeat experiments in Fig.3 using rotenone, antimycin and oligomycin. Mitochondrial ROS should be quantified by flowcytometry and data represented as histograms and mean of fluorescence for all the ETC inhibitors. Specifically, ROS quenching by NAC should be demonstrated. DNA damage and oxidation should also be assessed by phosphorylation of H2AX (FACS, IF) and 8-oxo-dG staining. Moreover, does GSH depletion through buthionine sulphoximine sensitize cells to OXPHOS inhibitors? PAR should be evaluated in response to rotenone, antimycin and oligomycin as in Fig. 3G.

We thank the reviewer for the comment and all the suggestions that help us clarify, at least in part, the effects of ROS in our experimental models.

First, we used Oligomycin to block ATP synthase. This inhibitor affects cell viability in a similar way to metformin or rotenone (results incorporated in the revised Fig. 3B).

Secondly, we have repeated ROS quantification by flow cytometry measuring intensity, and not percentage of positive cells. Using this new quantification method ROS levels present in p53/*Brca2*^{-/-} cells were similar to levels present in p53^{-/-} cells in basal conditions. Moreover, we confirmed higher ROS generation by metformin in p53/*Brca2*^{-/-} cells compared with p53^{-/-} cells after 48h of treatment. We have also evaluated ROS generation by the other ETC inhibitors: Rotenone (Rote), Oligomycin (Oligo) and Antimycin A (AA). Only inhibitors of the complex I and oligomycin cause an increase in ROS levels after 48h of treatment:

Lack of ROS generation by complex III inhibitors is surprising, given the well-known effects of these compounds in acute treatment.

The increase in ROS at 48h caused by the different inhibitors correlates with their effect on cell viability and the differences observed between p53^{-/-} and p53/Brca2^{-/-} cells: Cell viability is differentially affected by complex I inhibitors and oligomycin, whereas inhibitors of complex III have the same effect on both cell types. It is also important to stress that complex III inhibitors are very potent, and could be affecting viability at earlier time-points before a clear metabolic consequence could be observed, in contrast with metformin that allows a more adaptative response. Taken into account these results, we decided to eliminate the complex III inhibitors from the revised Fig. 3B, and include the oligomycin data. All these results also suggest that ROS generation could play, as suggested by the reviewer, an important role on the differential effect of complex I inhibitors and oligomycin on p53/Brca2^{-/-} versus p53^{-/-} viability. Moreover, when we add NAC to the cells and we measured ROS, NAC only partially blocked ROS generation induced by metformin in all the cells (20-30% decrease). We have also incubated cells in the presence of buthionine sulphoximine to deplete glutathione, but treatment did not alter metformin sensitivity. Given the complexity of the ROS data and the misunderstanding we did, we have decided to eliminate the data presented in the old Fig. 4E.

Whether or not ROS plays a role in the models presented here, our results indicate that the different metabolic capacities are the main players as suggested by the following: a) PAR activity in Brca2/p53^{-/-} cells in the presence of oxidative metabolism inhibitors such as rotenone blocked PAR activity in a similar manner to metformin, whereas, b) antimycin A did not affect PAR activity (when it affected cell viability), and c) oligomycin maximal effects on PAR activity:

Finally, PAR activity was independent of ROS, as agents that do not affect ROS but modify the metabolic capacities of cells, such as pyruvate or DCA, increased PAR activity (Fig. for the reviewer and new Fig. 3 and 4):

Also, as suggested by the reviewer, we have measured phosphorylation of histone H2AX after treatment with ETC inhibitors by western blot and by flow cytometry. Our results, incorporated in the new Figures 3 and Extended View Figure 4 in the revised manuscript, confirm that metformin treatment induces DNA damage in a similar manner to olaparib only in p53/Brca2 double mutated cells.

5) The authors describe that metformin treatment makes cells partially resistant to olaparib, as well as pyruvate. Does metformin exert a similar effect on olaparib IC50 in low glucose? It is possible that the resistance to olaparib can be simply mediated by an increased NAD⁺ regeneration through LDH in response to metformin and pyruvate. Can the authors measure lactate and assess NAD⁺/NADH ratio in this setting? If this hypothesis is correct a synergistic effect of metformin and olaparib should be expected in low glucose, a condition that can be explored in the clinic through fasting.

The ratio NAD^+/NADH decreased by metformin at 48h was caused by an increase in NADH levels, with small alterations on NAD^+ levels (old Fig. 4B and Sup. Fig. 11). Moreover, lactate was increased by metformin similarly in p53^{-/-} and p53/Brca2^{-/-} cells:

These data suggest that it is not the LDH activity that is responsible for the changes on olaparib sensitivity between p53^{-/-} and p53/Brca2^{-/-} cells.

We have also evaluated the reviewer hypothesis concerning the effect of glucose deprivation. We performed new experiments in media with low glucose (5 mM) and compared the results with those obtained in high glucose medium.

As shown, a decrease in glucose levels do not affect differential olaparib sensitivity between both cell lines. Moreover, the effect of metformin is identical at 5 mM and at 25 mM glucose. Secondly, a drop in glucose levels increase the olaparib sensitivity but only in glycolytic cells (p53^{-/-}). We reason that a glycolytic cell is more sensitive to changes in glucose availability than an OXPHOS-dependent cell, and in consequence, glycolytic cells become more sensitive to olaparib when glucose levels drop. Moreover, low glucose levels sensitize p53^{-/-} cells to metformin, increasing the

olaparib IC50 only at 5 mM but not at 25 mM glucose. These results confirmed again that changes on the metabolic profile have an impact on the sensitivity to olaparib.

6) Authors should attempt to validate the findings in vivo using PDXs.

We agree with the reviewer. To tackle the issue we are providing in the revised manuscript with a more suitable and complete set of experiments in vivo by using 2 additional models:

- Intraovarian injection of ID8 p53^{-/-} or p53/Brca2^{-/-} cells was used to grow tumors until they achieved a palpable size, with subsequent treatment with metformin or DMSO. Our results clearly indicate that p53/Brca2^{-/-} tumors are more sensitive to metformin than p53^{-/-} tumors, confirming the results obtained in cell culture conditions.

- Ovarian orthotopic PDX models, in which we compared a patient derived BRCA1 mutated ovarian tumor PDX with a patient WT BRCA1 ovarian tumor derived PDX. Again, the BRCA1 mutated PDX was more sensitive to metformin compared with the WT tumor, confirming our in vitro results. All these data have been incorporated in the new Figure 5 of the revised manuscript.

Minor points.

a) In Figure 1A, why do authors use expression as a surrogate of function Instead of mutational status?

The reviewer raises an important question. The somatic mutational signature originally named #3 (Alexandrov et al., Nature 2013) represents convergence on a functional defect (i.e., HR deficiency), that otherwise can be caused by different types of alterations, including germline and somatic BRCA1/2 mutations. However, following the reviewer's suggestion, we complied with BRCA1/2 mutation status of TCGA breast and ovarian cancer datasets from different sources (cBioPortal and TCGA accessible data; Yost et al., JNCI 2019; and Kraya et al., Clin Cancer Res 2019) and further analyzed the association with the KEGG Oxidative Phosphorylation pathway. In multivariate regression analyses including age at diagnosis and tumor stage, over-expression of this pathway was also detected significantly associated with BRCA1/2 mutations. These results are included in new panels in Figure 1A/B.

b) Differences in cell cycle between WT and HR deficient cells can be responsible for the described differences in OXPHOS and glycolysis. Although on page 8 authors claim there are no differences across models, differences are substantial in the workhorse model ID8 where double KO cells appear to be 38% slower than their trp53-deleted counterpart. Although these differences do not appear from Sup. Fig. 8A, the authors should provide cell cycle data and better comment on these results.

Differences on cell proliferation between ID8 p53^{-/-} and p53/Brca2^{-/-} cells exist, as confirmed by growing the same number of cells in culture plates and counting the number of cells at various times. A 30% lower number of cells was observed in exponentially growing p53/Brca2^{-/-} cells compared with p53^{-/-} cells at 48h and 72h.

The difference in cell number was similar to the changes observed by MTT or by crystal violet staining described in the manuscript and could be associated to the metabolic changes observed. We measured cell proliferative capacity in the different cell types studied and observed various scenarios as indicated in the table:

CELLS	GENETIC MODIFICATION	PROLIFERATION INDEX MODIFICATION	OXYGEN CONSUMPTION MODIFICATION
ID8 Mouse ovarian cancer	Trp53/Brca2 -/- versus Trp53 -/-	32% ↓	27% ↑
Mouse ovarian cancer (Orsulic Laboratory)	Trp53/Brca1 -/-, myc and Akt overexpression versus Trp53 -/-, myc and Akt overexpression	45% ↓	24% ↑
SKOV-3 resistant human ovarian cancer	Unknown	21% ↑	49% ↑
SKOV-3 BRCA2 deleted human ovarian cancer	TRP53/BRCA2 -/- versus TRP53 -/-	75% ↓	47% ↑
MEF mouse fibroblasts	Trp53/Tsc2 -/- versus Trp53	12% ↓	100% ↑

It is true that all the HRD cell models in culture present low proliferative rates compared with their respective non HRD models, probably caused by alterations in the cell cycle progression that we observed by flow cytometry analysis. Thus, we cannot discard that proliferation alterations could participate in the metabolic changes observed in our cell culture models. It is for this reason that we eliminate the sentence concerning the lack of effect of proliferation in our new version of the manuscript. But is also true that when we tried to confirm this in a more relevant human cancers setting (by Ki67 immunohistochemistry in the same ovarian WT or Brca2 tumoral patient samples analyzed in Fig. 1D and 1E showing significant changes on MCT4 and NDUFB2) we did not observe changes on tumoral proliferation in these patient samples (new Sup. Fig. 5). Therefore, we discarded changes on cell proliferation as the main reason for the metabolic changes observed in HRD tumors.

Additionally, even considering the possibility of an effect of proliferation on the metabolic adaptations found, it is important to stress that our work clearly show that by modulating metabolism we are able to modify PARP activity, for example using complex I inhibitors (lowering OXPHOS decreases PARP activity and decreases cell viability), with pyruvate or with DCA (increasing OXPHOS is sufficient to increase PARP activity).

c) Mitochondrial mass has not been quantified appropriately. A non-potential-dependent mitotracker (green) should be used and data quantified by FACS and represented as mean fluorescence. Poor quality immunoblots in Sup. Fig.8 should be replaced and immunofluorescence analysis for VDAC1 in ID8 should be considered.

The reviewer is right, so we reevaluated this part of the manuscript by measuring mitochondrial mass with a membrane potential-independent mitotracker and quantification by flow cytometry. Data is presented in the new Sup. Fig 6 suggesting a non-significant change on mitochondrial mass between *p53/Brca2* -/- and *p53* -/- cells.

d) Figure 2G, show protein level as well.

We tried to perform western blot experiments using antibodies against hexokinase and LDH, but blots were of low quality probably due to underperforming antibodies.

e) All the experiments are normalized on protein content? Can authors show data for cell number (DNA) at least for oxygen consumption.

All the experiments have been normalized by number of cells present at the time of the experiment. We have indicated this in the materials and methods section.

f) On page 4, the sentence "in this setting, inefficient ATP production due to incomplete oxidation of glucose is compensated by greater glucose consumption" is misleading. As reported in the cited references the Warburg effect is not the results of energetics rather the biosynthetic demand of proliferating cells.

We thank the reviewer for the revision, that is now properly stated in the new version of the manuscript.

g) The authors referred to Figure 5D,E instead of 5C,D.

We have corrected the reference in the new version of our manuscript.

2nd Editorial Decision

17th Mar 2020

Thank you for the submission of your revised manuscript to EMBO Molecular Medicine. We have now received the enclosed reports from the referees that were asked to re-assess it. As you will see the reviewers are now globally supportive and I am pleased to inform you that we will be able to accept your manuscript pending the following final amendments:

1) Please address the minor changes commented by the two referees, including the addition of data characterisation as requested by referee 1.

***** Reviewer's comments *****

Referee #1 (Comments on Novelty/Model System for Author):

see notes in the annotated rebuttal

Referee #1 (Remarks for Author):

please see notes in annotated rebuttal below. Marked with –

Referee #1

Major points

- Only one relevant murine model is used ID8 p53/BRCA2-/- . MCF7 and MDA-MB-231 are not good models for genomic instability comparison. Similarly, unless the mechanism of cisplatin-resistance in the SKOV-3-R pair is known, this is not a good Olaparib sensitive/responsive model system to use. Overall, the choice of good, well defined, isogenic model systems is a major limitation.

We agree with the reviewer that the work quality would improve with a better focus on only relevant models. For this reason, we have eliminated data from MCF7 and MDA-MB-231 from our article and we show mainly isogenic cell models. Thus, we have maintained all the data on ID8 p53 -/- or p53/Brca2 -/- cells and MEF p53 -/- or p53/TSC2 -/-. Moreover we have also included new data obtained on two new ovarian cancer cell models:

First, we have generated a new human SKOV-3 derived cell line in which we have deleted the BRCA2 gene by using a lentiviral GenCRISPR {trade mark, serif} gRNA/Cas9 lentiCRISPR v2 vector (GenScript, Piscataway, NJ) (SKOV-3 BRCA2-/- 1.2). SKOV-3 cells already present mutated p53. We generated a control SKOV-3-SC for the empty vector and SKOV-3-BRCA2-/- 1.2. Validation of the BRCA2 gene deletion was performed by sequencing, increase in the olaparib sensitivity, increase in the basal DNA damage (staining for phosphoHistone H2A.X Ser139) and lost of the ability to form RAD51 foci after olaparib treatment compared with the control cells (new Sup. Fig. 4).

– Have the authors carried out the basic validation of the new SKOV-3 model by (a) showing the sequence traces, (b) shown that the Brca2 protein is lost (c) show the metformin but also PARP inhibitor dose response curves

– The rad51 foci, as far as I can tell, are only shown as unconvincing images and not quantified

– For the column chart in EV figs 1 and 2 (and throughout the manuscript), better to show the actual data points as well as the median and deviation

– Did the authors look at metformin sensitivity in these new models ? Their logic somewhat rests on the minimal metformin sensitivity in Figure 3A.

We have also included new data concerning Brca1 deleted cells, obtained on new ovarian cancer cells obtained in Sandra Orsulic's laboratory (David Geffen School of Medicine at UCLA, Los Angeles, CA (USA)). These cells presented Trp53-/-, myc and Akt or Trp53-/-, Brca1-/-, myc and

Akt genetic alterations (Xing & Orsulic, Cancer Res. 2006) These cells were isolated from ovarian tumors after intraperitoneal injection of cells in nude mice.

In all these models with alterations in HR or genomic instability we have observed an increase in oxygen consumption and high PDH activation (new Extended View Figures 1 and 2 in the manuscript).

As suggested by the reviewers, we reserved the main figures of our manuscript for the ID8 based cell model and moved all the data from other cell lines to the Supplementary Data.

- A BRCA2 rescue experiment in the ID8 p53/BRCA2^{-/-} cell line is needed to confirm that loss of BRCA2 is responsible for the modest OXPHOS differences observed.

Unfortunately, we failed to recover the expression of Brca2 on ID8 p53/Brca2^{-/-} cells. As an alternative experiment, we obtained MDA-MB-436 from a collaborator overexpressing or not Brca1 by retroviral expression. The cell line MDA-MB-436 presents the 5396 + 1G>A mutation in the splice donor site of exon 20, classified as pathogenic mutations, and accompanied by loss of the other BRCA1 allele (Eldstrot et al. Cancer Res. 2006; 66(1): 41-5). Cells overexpressing the WT form of BRCA1 presented less sensitivity to olaparib compared with the normal MDA-MB-436. Oxygen consumption in these cells did not unveiled differences between Brca1^{-/-} and Brca1 overexpressing cells.

These results suggest that changes on OXPHOS could be caused, not by direct genetic regulation by Brca proteins or the complex that they form, but by the increase in chronic DNA damage, as suggested by experiments of other authors using normal untransformed cells (Brace et al, 2016, NPJ Aging Mech Dis 2: 16022).

– This is interesting, but the title of the manuscript still implies that a homologous recombination defect is the cause of the main phenotypes (the above result suggests not). As the second reviewer rightly points out the authors interchange genomic instability/homologous recombination defects and Brca gene defects as though these are all interchangeable descriptions of the same thing. They are not. I think it's important for the authors to decide which of these is actually causing the metabolic phenotypes. For example, transient knockdown of Brca in a diploid cell line (eg CAL51) will cause a HR defect, PARPi sensitivity etc, but in the short term will not cause large-scale genomic instability (prolonged silencing will however). Likewise, reversion mutations in BRCA genes restore BRCA gene function, restore HR but do not reverse large-scale genomic rearrangements. I still think the choice of models and the way the data are interpreted from these is confusion/inconsistent etc.

- Figure 2A and B - Is the observed increase in O₂ consumption due to proliferation alterations or genomic instability? The paper states that proliferation is not the cause but authors do not present the cell line proliferation data.

Differences on cell proliferation between ID8 p53^{-/-} and p53/Brca2^{-/-} cells exist, as confirmed by growing the same number of cells in culture plates and counting the number of cells at various times. A 30% lower number of cells was observed in exponentially growing p53/Brca2^{-/-} cells compared with p53^{-/-} cells at 48h and 72h.

The difference in cell number was similar to the changes observed by MTT or by crystal violet staining described in the manuscript and could be associated to the metabolic changes observed. We measured cell proliferative capacity in the different cell types studied and observed various scenarios as indicated in the table:

CELLS GENETIC MODIFICATION PROLIFERATION INDEX MODIFICATION OXIGEN CONSUMPTION MODIFICATION

ID8 Mouse ovarian cancer Trp53/Brca2^{-/-} versus Trp53^{-/-} 32% ↓ 27% ↑

Mouse ovarian cancer

(Orsulic Laboratory) Trp53/Brca1^{-/-}, myc and Akt overexpression versus

Trp53^{-/-}, myc and Akt overexpression 45% ↓ 24% ↑

SKOV-3 resistant human ovarian cancer Unknown 21% ↑ 49% ↑

SKOV-3 BRCA2 deleted human ovarian cancer TRP53/BRCA2^{-/-} versus TRP53^{-/-} 75% ↓ 47% ↑

MEF mouse fibroblasts Trp53/Tsc2^{-/-} versus Trp53 12% ↓ 100% ↑

It is true that all the HRD cell models in culture present low proliferative rates compared with their respective non HRD models, probably caused by alterations in the cell cycle progression that we

observed by flow cytometry analysis. Thus, we cannot discard that proliferation alterations could participate in the metabolic changes observed in our cell culture models. It is for this reason that we eliminate the sentence concerning the lack of effect of proliferation in our new version of the manuscript. But it is also true that when we tried to confirm this in a more relevant human cancers setting (by Ki67 immunohistochemistry in the same ovarian WT or Brca2 tumoral patient samples analyzed in Fig. 1D and 1E showing significant changes on MCT4 and NDUFV2) we did not observe changes on tumoral proliferation in these patient samples (new Sup. Fig. 1B). Therefore, we discarded changes on cell proliferation as the main reason for the metabolic changes observed in HRD tumors.

Additionally, even considering the possibility of an effect of proliferation on the metabolic adaptations found, it is important to stress that our work clearly shows that by modulating metabolism we are able to modify PARP activity, for example using complex I inhibitors (lowering OXPHOS decreases PARP activity and decreases cell viability), with pyruvate or with DCA (increasing OXPHOS is sufficient to increase PARP activity).

- Provide some sort of control for glucose/lactate/glycolytic assays / provide some context to help explain the significance of the observed decrease in activity in the ID8 p53/BRCA2^{-/-} model. The significant changes in glycolytic flux or lactate production observed in p53/Brca2^{-/-}; 18% decrease in glycolytic flux and a 15% decrease in lactate production are consistent with the amplitude of the global metabolic changes described, probably as a consequence of an adaptation to the high requirements of ATP and NAD⁺ in cells with a high DNA repair activity.

These changes are of similar magnitude to those described in the literature by other metabolic effectors in tumoral cells. For example, the transcription factor BACH1 controls metabolic enzymes and ETC genes expression decreasing glucose utilization in the TCA and OXPHOS metabolism. Decreasing BACH1 expression shifts breast cancer cells to a more OXPHOS dependent metabolism, with quantitative changes that are equivalent to our p53/Brca2^{-/-} cells. Decrease in BACH1 levels only causes a drop in of 15% in lactate level, for example (Lee et al., Nature 568: 254, 2019, Extended Data Fig. 2B).

- If the authors are suggesting upregulation of TIGAR is responsible for inhibiting glycolysis in the p53/BRCA2^{-/-} model, knocking down TIGAR and examining a reversion of the metabolic changes observed seems critical

In order to explore a possible role of TIGAR in our experimental setting, we modified expression of TIGAR by siRNA treatment (decreasing TIGAR levels) or by stable overexpression (doubling TIGAR levels), but we did not observe changes in OXPHOS in any of these cells. Moreover, when we analyzed TIGAR gene expression and OXPHOS genes expression data from the breast and ovarian cancer studies of the TCGA, we did not find a significant correlation between them. In all, these results suggest to us that we should consider changes in TIGAR expression more as a consequence of a metabolic adaptation, not as the cause of the metabolic changes observed. We have obtained preliminary data suggesting that the changes on OXPHOS proteins and other metabolic enzymes consequence of DNA damage could be induced by transcription factors such as BACH1, a protein that, as mentioned before, controls OXPHOS gene expression.

– It would be good for the authors to point to the changes in the manuscript where they have modified their conclusions accordingly

- There are some leaps of faith made - why were MCT4 and NDUFV2 examined in figure 1? Complex I is a large protein complex made up of 45/46 different subunits. Why did they look at this subunit in particular, was it enriched in the GSEA analysis? Did they look at other Complex I subunits?

Unfortunately, we failed to properly explain this dataset in the manuscript, and hence the spot-on criticisms by the reviewer and others. We chose NDUFV2 subunit of the complex I for two reasons:

- First, because this subunit was one of the genes that were increased in the HR defective associated S3⁺ signature that contribute to the GSEA analysis.

- Second, antibodies against this protein were validated in immunohistochemistry analysis by other authors.

Concerning MCT4, the election was based in its role on lactate extrusion from tumoral cells described in the bibliography and in the availability of high-quality antibodies to perform immunohistochemical analysis.

– It would be good for the authors to point to the changes in the manuscript where they have

described the above arguments

- No substantial *in vivo* validation of observed biology - the analysis of the PDX data seems limited. We agree with the reviewer. To tackle the issue we are providing in the revised manuscript with a more suitable and complete set of experiments *in vivo* by using 2 additional models:
 - Intraovarian injection of ID8 p53^{-/-} or p53/Brca2^{-/-} cells was used to grow tumors until they achieved a palpable size, with subsequent treatment with metformin or vehicle. Our results clearly indicate that p53/Brca2^{-/-} tumors are more sensitive to metformin than p53^{-/-} tumors, confirming the results obtained in cell culture conditions.
 - Ovarian orthotopic PDX models, in which we compared a patient derived BRCA1 mutated ovarian tumor PDX with a patient WT BRCA1 ovarian tumor derived PDX. Again, the BRCA1 mutated PDX was more sensitive to metformin compared with the WT tumor, confirming our *in vitro* results. All these data have been incorporated in the new Figure 5 of the revised manuscript.

- It would help the reader if the authors could show the absolute tumor volume and fold change in tumor volume over time (as is commonly done). It is not clear from the methods whether tumor size at the start of treatment was consistent between different animals. Also in the methods, the authors don't note that tumors were palpable at the start of treatment (although this detail is given above).

Minor points

- Figure 5D - PARP trapping western; cytoplasmic fractionation loading controls needed to rule out contamination in nuclear fraction.

As suggested by the reviewer, in the new version of new Figure 6 we incorporated immunoblot for tubulin as a cytoplasmic marker. As shown in the figure, the amount of tubulin decreases with increased astringency in the wash steps, until finally disappears in fraction 3, where we detected the presence of PARP1.

- Supplementary Figure 4A - quantification of mitotracker would make comparison easier and scale bars for reference

The reviewer is right, so we reevaluated this part of the manuscript by measuring mitochondrial mass with a membrane potential-independent mitotracker and quantification by flow cytometry. Data is presented in the new Sup. Fig 6 suggesting a non-significant change on mitochondrial mass between p53/Brca2^{-/-} and p53^{-/-} cells.

- Supplementary Figure 4B-D - Vinculin wrote out in full for consistency

The reviewer is right and, as indicated in the previous point, we modified the quantification and interpretation of the data in the revised manuscript.

- Supplementary Figure 4D - looks like VDAC is more highly expressed in SKOV-3 compared to SKOV3-R and paper states unaffected mitochondria mass is seen in models ?

The reviewer is again right, but we have no explanation for these changes.

- Supplementary Figure 5B - GLUT1 appears glycosylated in HR competent models but this is not commented on ?

The differences in the glycosylation level observed have not been studied in detail, but it is possible that differences in N-acetyl glucosamine provision could explain these differences.

- Figure 6B - spelling mistake Oxigen -> Oxygen

We corrected this mistake in the manuscript.

Referee #2 (Remarks for Author):

The manuscript by Lahiguera et al. identifies OXPHOS as a new vulnerability in tumors characterized by deficient homologous recombination (HR). Mechanistically, the dependency on mitochondrial respiration of HR deficient cells is mediated by the increased demand of NAD⁺ and ATP necessary to sustain PARP activity. Furthermore, shifting metabolism from oxidative to glycolytic would reduce sensitivity to PARP inhibitors.

Although this is a very interesting study there are some key points the authors have to address before the manuscript is suitable for publication.

1) It is not clear if the phenotype described by the authors is a general feature of genomically unstable tumors or a specific characteristic of HR deficient cells, as title and data suggest. The authors should be consistent throughout the manuscript and avoid to jump from one to another. We completely agree with the reviewer on this. Even though we hypothesize that the conclusions of the present work will probably concern non HR-defective, genomic instable cells, our data is limited to HR-defective models, and so our interpretation will focus mainly on those datasets.

2) A major concern is about the models used. Besides ID8, that is the most reliable and controlled one since cells harbor specific genomic events in a well-defined genomic background, all the other models are questionable and poorly matched. As an example, how the authors take into consideration genomic differences between MCF7 and MDA-MB-231. I would suggest to focus the manuscript on ID8 model only and move all the other data to supplementary figures. It would be ideal to have another isogenic model, maybe breast, and repeat at least some key experiments. We agree with the reviewer that the work quality would improve with a better focus on only relevant models. For this reason, we have eliminated data from MCF7 and MDA-MB-231 from our article and we show mainly isogenic cell models. Thus, we have maintained all the data on ID8 p53^{-/-} or p53/Brca2^{-/-} cells and MEF p53^{-/-} or p53/TSC2^{-/-}. Moreover we have also included new data obtained on two new ovarian cancer cell models:

First, we have generated a new human SKOV-3 derived cell line in which we have deleted the BRCA2 gene by using a lentiviral GenCRISPR[®] gRNA/Cas9 lentiCRISPR v2 vector (GenScript, Piscataway, NJ) (SKOV-3 BRCA2^{-/-} 1.2). SKOV-3 cells already present mutated p53. We generated a control SKOV-3-SC for the empty vector and SKOV-3-BRCA2^{-/-} 1.2. Validation of the BRCA2 gene deletion was performed by sequencing, increase in the olaparib sensitivity, increase in the basal DNA damage (staining for phosphoHistone H2A.X Ser139) and lost of the ability to form RAD51 foci after olaparib treatment compared with the control cells (new Sup. Fig. 4).

We have also included new data concerning Brca1 deleted cells obtained on new ovarian cancer cells obtained in Sandra Orsulic's laboratory (David Geffen School of Medicine at UCLA, Los Angeles, CA (USA)). These cells presented Trp53^{-/-}, myc and Akt or Trp53^{-/-}, Brca1^{-/-}, myc and Akt genetic alterations (Xing & Orsulic, Cancer Res. 2006) These cells were isolated from ovarian tumors after intraperitoneal injection of cells in nude mice.

In all these models with alterations in HR or genomic instability we have observed an increase in oxygen consumption and high PDH activation (new Extended View Figures 1 and 2 in the manuscript).

As suggested by the reviewers, we reserved the main figures of our manuscript for the ID8 based cell model and moved all the data from other cell lines to the Supplementary Data.

3) Instead of partial assessments of metabolic pathways, an exhaustive representation of the metabolic changes in HR deficient cells would be better captured by metabolomics and isotopic tracing. Can the authors consider these approaches?

We thank the reviewer for the comment and suggestion. Following her/his recommendation we have performed a metabolomic study using ¹³C-glucose as a tracer. Results obtained (incorporated in the new Fig. 2F and new Fig. 4A) confirmed our initial observations indicating a higher flux of carbons towards the Krebs cycle in the p53/Brca2^{-/-} cells. Moreover, incubation in the presence of metformin causes a blockade in OXPHOS metabolism of both p53/Brca2^{-/-} and p53^{-/-} cells.

4) Another critical point not addressed by the authors is why HR defective cells are not sensitive to other ETC inhibitors. If continuous supplementation of NAD⁺ and ATP is necessary to sustain PARP activity, as hypothesized by the authors, the same effect would be expected irrespective to which ETC complex is targeted. Did authors attempt to inhibit complex V? Are ROS really not involved in this phenotype? One would expect that increased ROS further damage DNA making HR deficient cells exquisitely sensitive to ETC inhibitors. Because a key point, authors have to repeat experiments in Fig.3 using rotenone, antimycin and oligomycin. Mitochondrial ROS should be quantified by flowcytometry and data represented as histograms and mean of fluorescence for all the ETC inhibitors. Specifically, ROS quenching by NAC should be demonstrated. DNA damage and oxidation should also be assessed by phosphorylation of H2AX (FACS, IF) and 8-oxo-dG staining. Moreover, does GSH depletion through buthionine sulphoximine sensitize cells to OXPHOS inhibitors? PAR should be evaluated in response to rotenone, antimycin and oligomycin as in Fig. 3G.

We thank the reviewer for the comment and all the suggestions that help us clarify, at least in part,

the effects of ROS in our experimental models.

First, we used Oligomycin to block ATP synthase. This inhibitor affects cell viability in a similar way to metformin or rotenone (results incorporated in the revised Fig. 3B).

Secondly, we have repeated ROS quantification by flow cytometry measuring intensity, and not percentage of positive cells. Using this new quantification method ROS levels present in p53/Brca2 $-/-$ cells were similar to levels present in p53 $-/-$ cells in basal conditions. Moreover, we confirmed higher ROS generation by metformin in p53/Brca2 $-/-$ cells compared with p53 $-/-$ cells after 48h of treatment. We have also evaluated ROS generation by the other ETC inhibitors: Rotenone (Rote), Oligomycin (Oligo) and Antimycin A (AA). Only inhibitors of the complex I and oligomycin cause an increase in ROS levels after 48h of treatment:

Lack of ROS generation by complex III inhibitors is surprising, given the well-known effects of these compounds in acute treatment.

The increase in ROS at 48h caused by the different inhibitors correlates with their effect on cell viability and the differences observed between p53 $-/-$ and p53/Brca2 $-/-$ cells: Cell viability is differentially affected by complex I inhibitors and oligomycin, whereas inhibitors of complex III have the same effect on both cell types. It is also important to stress that complex III inhibitors are very potent, and could be affecting viability at earlier time-points before a clear metabolic consequence could be observed, in contrast with metformin that allows a more adaptative response. Taken into account these results, we decided to eliminate the complex III inhibitors from the revised Fig. 3B, and include the oligomycin data. All these results also suggest that ROS generation could play, as suggested by the reviewer, an important role on the differential effect of complex I inhibitors and oligomycin on p53/Brca2 $-/-$ versus p53 $-/-$ viability. Moreover, when we add NAC to the cells and we measured ROS, NAC only partially blocked ROS generation induced by metformin in all the cells (20-30% decrease). We have also incubated cells in the presence of buthionine sulfoximine to deplete glutathione, but treatment did not alter metformin sensitivity. Given the complexity of the ROS data and the misunderstanding we did, we have decided to eliminate the data presented in the old Fig. 4E.

Whether or not ROS plays a role in the models presented here, our results indicate that the different metabolic capacities are the main players as suggested by the following: a) PAR activity in Brca2/p53 $-/-$ cells in the presence of oxidative metabolism inhibitors such as rotenone blocked PAR activity in a similar manner to metformin, whereas, b) antimycin A did not affect PAR activity (when it affected cell viability), and c) oligomycin maximal effects on PAR activity:

Finally, PAR activity was independent of ROS, as agents that do not affect ROS but modify the metabolic capacities of cells, such as pyruvate or DCA, increased PAR activity (Fig. for the reviewer and new Fig. 3 and 4):

Also, as suggested by the reviewer, we have measured phosphorylation of histone H2AX after treatment with ETC inhibitors by western blot and by flow cytometry. Our results, incorporated in the new Figures 3 and Extended View Figure 4 in the revised manuscript, confirm that metformin treatment induces DNA damage in a similar manner to olaparib only in p53/Brca2 double mutated cells.

5) The authors describe that metformin treatment makes cells partially resistant to olaparib, as well as pyruvate. Does metformin exert a similar effect on olaparib IC₅₀ in low glucose? It is possible that the resistance to olaparib can be simply mediated by an increased NAD⁺ regeneration through LDH in response to metformin and pyruvate. Can the authors measure lactate and assess NAD⁺/NADH ratio in this setting? If this hypothesis is correct a synergistic effect of metformin and olaparib should be expected in low glucose, a condition that can be explored in the clinic through fasting.

The ratio NAD⁺/NADH decreased by metformin at 48h was caused by an increase in NADH levels, with small alterations on NAD⁺ levels (old Fig. 4B and Sup. Fig. 11).

Moreover, lactate was increased by metformin similarly in p53 $-/-$ and p53/Brca2 $-/-$ cells:

These data suggest that it is not the LDH activity that is responsible for the changes on olaparib sensitivity between p53 $-/-$ and p53/Brca2 $-/-$ cells.

We have also evaluated the reviewer hypothesis concerning the effect of glucose deprivation. We performed new experiments in media with low glucose (5 mM) and compared the results with those obtained in high glucose medium.

As shown, a decrease in glucose levels do not affect differential olaparib sensitivity between both cell lines. Moreover, the effect of metformin is identical at 5 mM and at 25 mM glucose. Secondly, a drop in glucose levels increase the olaparib sensitivity but only in glycolytic cells (p53^{-/-}). We reason that a glycolytic cell is more sensitive to changes in glucose availability than an OXPHOS-dependent cell, and in consequence, glycolytic cells become more sensitive to olaparib when glucose levels drop. Moreover, low glucose levels sensitize p53^{-/-} cells to metformin, increasing the olaparib IC50 only at 5 mM but not at 25 mM glucose. These results confirmed again that changes on the metabolic profile have an impact on the sensitivity to olaparib.

6) Authors should attempt to validate the findings in vivo using PDXs.

We agree with the reviewer. To tackle the issue we are providing in the revised manuscript with a more suitable and complete set of experiments in vivo by using 2 additional models:

- Intraovarian injection of ID8 p53^{-/-} or p53/Brca2^{-/-} cells was used to grow tumors until they achieved a palpable size, with subsequent treatment with metformin or DMSO. Our results clearly indicate that p53/Brca2^{-/-} tumors are more sensitive to metformin than p53^{-/-} tumors, confirming the results obtained in cell culture conditions.

- Ovarian orthotopic PDX models, in which we compared a patient derived BRCA1 mutated ovarian tumor PDX with a patient WT BRCA1 ovarian tumor derived PDX. Again, the BRCA1 mutated PDX was more sensitive to metformin compared with the WT tumor, confirming our in vitro results. All these data have been incorporated in the new Figure 5 of the revised manuscript.

Minor points.

a) In Figure 1A, why do authors use expression as a surrogate of function Instead of mutational status?

The reviewer raises an important question. The somatic mutational signature originally named #3 (Alexandrov et al., Nature 2013) represents convergence on a functional defect (i.e., HR deficiency), that otherwise can be caused by different types of alterations, including germline and somatic BRCA1/2 mutations. However, following the reviewer's suggestion, we complied with BRCA1/2 mutation status of TCGA breast and ovarian cancer datasets from different sources (cBioPortal and TCGA accessible data; Yost et al., JNCI 2019; and Kraya et al., Clin Cancer Res 2019) and further analyzed the association with the KEGG Oxidative Phosphorylation pathway. In multivariate regression analyses including age at diagnosis and tumor stage, over-expression of this pathway was also detected significantly associated with BRCA1/2 mutations. These results are included in new panels in Figure 1A/B.

b) Differences in cell cycle between WT and HR deficient cells can be responsible for the described differences in OXPHOS and glycolysis. Although on page 8 authors claim there are no differences across models, differences are substantial in the workhorse model ID8 where double KO cells appear to be 38% slower than their trp53-deleted counterpart. Although these differences do not appear from Sup. Fig. 8A, the authors should provide cell cycle data and better comment on these results.

Differences on cell proliferation between ID8 p53^{-/-} and p53/Brca2^{-/-} cells exist, as confirmed by growing the same number of cells in culture plates and counting the number of cells at various times. A 30% lower number of cells was observed in exponentially growing p53/Brca2^{-/-} cells compared with p53^{-/-} cells at 48h and 72h.

The difference in cell number was similar to the changes observed by MTT or by crystal violet staining described in the manuscript and could be associated to the metabolic changes observed. We measured cell proliferative capacity in the different cell types studied and observed various scenarios as indicated in the table:

CELLS GENETIC MODIFICATION PROLIFERATION INDEX MODIFICATION OXIGEN CONSUMPTION MODIFICATION

ID8 Mouse ovarian cancer Trp53/Brca2^{-/-} versus Trp53^{-/-} 32% ↓ 27% ↑

Mouse ovarian cancer

(Orsulic Laboratory) Trp53/Brca1^{-/-}, myc and Akt overexpression versus

Trp53^{-/-}, myc and Akt overexpression 45% ↓ 24% ↑

SKOV-3 resistant human ovarian cancer Unknown 21% ↑ 49% ↑

SKOV-3 BRCA2 deleted human ovarian cancer TRP53/BRCA2^{-/-} versus TRP53^{-/-} 75% ↓ 47% ↑

MEF mouse fibroblasts Trp53/Tsc2^{-/-} versus Trp53 12% ↓ 100% ↑

It is true that all the HRD cell models in culture present low proliferative rates compared with their respective non HRD models, probably caused by alterations in the cell cycle progression that we observed by flow cytometry analysis. Thus, we cannot discard that proliferation alterations could participate in the metabolic changes observed in our cell culture models. It is for this reason that we eliminate the sentence concerning the lack of effect of proliferation in our new version of the manuscript. But it is also true that when we tried to confirm this in a more relevant human cancers setting (by Ki67 immunohistochemistry in the same ovarian WT or Brca2 tumoral patient samples analyzed in Fig. 1D and 1E showing significant changes on MCT4 and NDUFV2) we did not observe changes on tumoral proliferation in these patient samples (new Sup. Fig. 5). Therefore, we discarded changes on cell proliferation as the main reason for the metabolic changes observed in HRD tumors.

Additionally, even considering the possibility of an effect of proliferation on the metabolic adaptations found, it is important to stress that our work clearly show that by modulating metabolism we are able to modify PARP activity, for example using complex I inhibitors (lowering OXPHOS decreases PARP activity and decreases cell viability), with pyruvate or with DCA (increasing OXPHOS is sufficient to increase PARP activity).

c) Mitochondrial mass has not been quantified appropriately. A non-potential-dependent mitotracker (green) should be used and data quantified by FACS and represented as mean fluorescence. Poor quality immunoblots in Sup. Fig.8 should be replaced and immunofluorescence analysis for VDAC1 in ID8 should be considered.

The reviewer is right, so we reevaluated this part of the manuscript by measuring mitochondrial mass with a membrane potential-independent mitotracker and quantification by flow cytometry. Data is presented in the new Sup. Fig 6 suggesting a non-significant change on mitochondrial mass between p53/Brca2 $-/-$ and p53 $-/-$ cells.

d) Figure 2G, show protein level as well.

We tried to perform western blot experiments using antibodies against hexokinase and LDH, but blots were of low quality probably due to underperforming antibodies.

e) All the experiments are normalized on protein content? Can authors show data for cell number (DNA) at least for oxygen consumption.

All the experiments have been normalized by number of cells present at the time of the experiment. We have indicated this in the materials and methods section.

f) On page 4, the sentence "in this setting, inefficient ATP production due to incomplete oxidation of glucose is compensated by greater glucose consumption" is misleading. As reported in the cited references the Warburg effect is not the results of energetics rather the biosynthetic demand of proliferating cells.

We thank the reviewer for the revision, that is now properly stated in the new version of the manuscript.

g) The authors referred to Figure 5D,E instead of 5C,D.

We have corrected the reference in the new version of our manuscript.

Referee #2 (Remarks for Author):

The authors addressed the majority of my concerns. I recommend the manuscript for publication upon minor changes.

On page 14 the sentence "As glucose is finely regulated to either feed the OXPHOS or the glycolytic pathway..." should be rephrased since glycolysis and OXPHOS are sequential not alternative pathways. Maybe authors meant aerobic glycolysis and lactate production.

On page 16. The sentence "Collectively, these data indicated that OXPHOS is increased in Trp53/Brca2-deleted cells secondary to, at least, increased glucose mitochondrial metabolism" should be rephrased as "increased contribution of glucose to mitochondrial metabolism".

On page 17. Authors should rephrase "The metformin mechanism affected cell viability and the cell

cycle of Trp53-deleted cells and Trp53/Brca2-deleted cells was disrupted (Fig. EV3A), with no increase in apoptosis (Fig. EV3B)" since not correct and confusing.

On page 20, authors meant unsensitive not sensitive (...carbon flux and metabolite pools were sensitive to the presence of Brca2...)

On page 23 replace "progression" with "growth". Also, in figure 5, can authors show histology and Ki67 of treated and untreated tumors?

Labelled TCA intermediates in figure 2f are difficult to read, please consider to replace with a panel similar to Fig.4a

2nd Revision - authors' response

5th Apr 2020

Referee #1

Ø Have the authors carried out the basic validation of the new SKOV-3 model by (a) showing the sequence traces, (b) shown that the Brca2 protein is lost (c) show the metformin but also PARP inhibitor dose response curves

Yes, we carried out the validation of our SKOV-3-BRCA2-/- 1.2 model by:

a) Sequencing the strip of the BRCA2 gene targeted by the GenCRISPR™ gRNA/Cas9 lentiCRISPR v2 vector: CTGTCTACCTGACCAATCGA. The sequence is altered from the nt targeted by the vector begins. We generated a new version of Appendix Figure 4 including the limited BRCA2 gene sequence.

b) Our efforts to obtain a credible western blot of the BRCA2 protein failed. It was for this reason that we validated BRCA2 loss by other methods, such as olaparib sensitivity, phosphoHistone H2A.X Ser139 levels (as a measure of the basal DNA damage) and loss of the ability to form RAD51 foci after olaparib treatment. These experiments validate loss of BRCA2 expression and have been previously used by other groups in a similar context (see for example Walton et al., Cancer Res. 76: 6118, 2016).

c) The increase in olaparib sensitivity was measured by measuring IC50; 2.8 mM for SKOV-3-BRCA2-/- 1.2 compared with 6.9 mM for control SKOV-3-SC.

All these data have been incorporated in the new version of our manuscript (Materials and Methods section and in the Appendix Figure 4).

Ø The rad51 foci, as far as I can tell, are only shown as unconvincing images and not quantified
The Illustrator figure is of much higher quality than the PDF showed. We hope that the quality of the images for Rad51 will be maintained in the final version of the article. No quantification of the images was performed because the result was binary (all or none), thus no labeling was noted in the absence of olaparib, loci were found in the presence of olaparib, and a total absence of loci were found in the clone SKOV-3-BRCA2-/- 1.2 in the absence or presence of olaparib.

Ø For the column chart in EV figs 1 and 2 (and throughout the manuscript), better to show the actual data points as well as the median and deviation

Throughout all the manuscript we have reserved the use of individual points for experiments with data from patients or mice (Fig 1D, E and F, Fig. 5, Fig. 7, etc). In the case of the experiments carried out with cells and that we consider more as independent replicates, we have preferred the use of graphs with columns representing the mean +/- SEM.

Ø Did the authors look at metformin sensitivity in these new models? Their logic somewhat rests on the minimal metformin sensitivity in Figure 3A.

We have measured metformin sensitivity for the SKOV-3-BRCA2-/- 1.2 model. Results also indicate a 33% decrease of the metformin IC50 in the SKOV-3-BRCA2-/- 1.2 cells (8.5 mM) compared with control SKOV-3-SC cells (12.6 mM):

We have not confirmed these results by using other ETC inhibitors and we decided do not include them in the manuscript.

- A BRCA2 rescue experiment in the ID8 p53/BRCA2^{-/-} cell line is needed to confirm that loss of BRCA2 is responsible for the modest OXPHOS differences observed.

Unfortunately, we failed to recover the expression of Brca2 on ID8 p53/Brca2^{-/-} cells. As an alternative experiment, we obtained MDA-MB-436 from a collaborator overexpressing or not Brca1 by retroviral expression. The cell line MDA-MB-436 presents the 5396 + 1G>A mutation in the splice donor site of exon 20, classified as pathogenic mutations, and accompanied by loss of the other BRCA1 allele (Eldstrot et al. Cancer Res. 2006; 66(1): 41-5). Cells overexpressing the WT form of BRCA1 presented less sensitivity to olaparib compared with the normal MDA-MB-436. Oxygen consumption in these cells did not unveiled differences between Brca1^{-/-} and Brca1 overexpressing cells.

These results suggest that changes on OXPHOS could be caused, not by direct genetic regulation by Brca proteins or the complex that they form, but by the increase in chronic DNA damage, as suggested by experiments of other authors using normal untransformed cells (Brace et al, 2016, NPJ Aging Mech Dis 2: 16022).

Ø This is interesting, but the title of the manuscript still implies that a homologous recombination defect is the cause of the main phenotypes (the above result suggests not). As the second reviewer rightly points out the authors interchange genomic instability/homologous recombination defects and Brca gene defects as though these are all interchangeable descriptions of the same thing. They are not. I think it's important for the authors to decide which of these is actually causing the metabolic phenotypes. For example, transient knockdown of Brca in a diploid cell line (eg CAL51) will cause a HR defect, PARPi sensitivity etc, but in the short term will not cause large-scale genomic instability (prolonged silencing will however). Likewise, reversion mutations in BRCA genes restore BRCA gene function, restore HR but do not reverse large-scale genomic rearrangements. I still think the choice of models and the way the data are interpreted from these is confusion/inconsistent etc.

As we previously exposed, our data is limited to HR-defective models, and so our interpretation will focus mainly on those datasets. For this reason, and to be more consistent, we changed our conclusions throughout the manuscript to limit our focus on HR-defective cells, as suggested by the reviewer. It is also important to stress that our data concern not only cell culture models, but also data obtained in real patient tumors, results that confirm our conclusions. The other criticism raised by the reviewer on the cause for the metabolic phenotype observed (reliance on OXPHOS metabolism) is interesting but far from the objectives and limitations of our study. If the increase in OXPHOS metabolism is caused by

large-scale genomic instability or directly by defects on the HR machinery requires further research. For example, studies focused on identify the transcription factors and signalling pathways involved in this control. Rescue experiments with BRCA2 as suggested by the reviewer, but also of rescue with BRCA1 in a BRCA2-defective background, will contribute to clarify this point in the future.

- If the authors are suggesting upregulation of TIGAR is responsible for inhibiting glycolysis in the p53/BRCA2^{-/-} model, knocking down TIGAR and examining a reversion of the metabolic changes observed seems critical

Ø It would be good for the authors to point to the changes in the manuscript where they have modified their conclusions accordingly

We eliminate the sentence on the involvement of TIGAR (“that promotes OXPHOS when overexpressed in breast cancer cells”) since it did promote the idea that this protein has a role in the context of our models. We have not indicated any further role for TIGAR in the rest of the manuscript.

- There are some leaps of faith made - why were MCT4 and NDUFB2 examined in figure 1 ? Complex I is a large protein complex made up of 45/46 different subunits. Why did they look at this subunit in particular, was it enriched in the GSEA analysis? Did they look at other Complex I subunits?

Ø It would be good for the authors to point to the changes in the manuscript where they have described the above arguments

We have “spell-out” the reason for NDUFB2 use, as follows: “because this subunit was one of the genes that were increased in the HR defective associated S3+ signature that contribute to the GSEA analysis” in the appropriate section of the manuscript.

No substantial in vivo validation of observed biology - the analysis of the PDX data seems limited

Ø It would help the reader if the authors could show the absolute tumor volume and fold change in tumor volume over time (as is commonly done).

It is important to stress that the in vivo model shown is not a subcutaneous model (in which it is possible to follow the tumour volume size over time) but an orthotopic ovary model, in which the tumour has been implanted in the ovary of the mice. For this reason, it is not possible to measure the tumour volume or size over time except at the end-point of the experiment. It is possible though, to assess tumoral growth and the approximate size of the tumour by palpation.

It is not clear from the methods whether tumor size at the start of treatment was consistent between different animals. Also in the methods, the authors don't note that tumors were palpable at the start of treatment (although this detail is given above).

As indicated in the manuscript, when a palpable intra-abdominal mass was detected, animals were randomized into two groups and metformin treatment started. Randomization ensures distribution of animals with similar tumor volumes in the two groups. We clarify this point in the manuscript.

Referee #2 (Remarks for Author):

The authors addressed the majority of my concerns. I recommend the manuscript for publication upon minor changes.

On page 14 the sentence "As glucose is finely regulated to either feed the OXPHOS or the glycolytic Pathway..." should be rephrased since glycolysis and OXPHOS are sequential not alternative pathways. Maybe authors meant aerobic glycolysis and lactate production.

The reviewer is right, we rephrased the sentence to indicate the use of glucose to OXPHOS or to lactate production.

On page 16. The sentence "Collectively, these data indicated that OXPHOS is increased in Trp53/Brca2-deleted cells secondary to, at least, increased glucose mitochondrial metabolism" should be rephrased as "increased contribution of glucose to mitochondrial metabolism".

As suggested by the reviewer we rephrased the sentence.

On page 17. Authors should rephrase "The metformin mechanism affected cell viability and the cell cycle of Trp53-deleted cells and Trp53/Brca2-deleted cells was disrupted (Fig. EV3A), with no increase in apoptosis (Fig. EV3B)" since not correct and confusing.

We rephrased as indicated.

On page 20, authors meant insensitive not sensitive (...carbon flux and metabolite pools were sensitive to the presence of Brca2...)

The reviewer is right, we changed the word on the new version of the manuscript.

On page 23 replace "progression" with "growth". Also, in figure 5, can authors show histology and Ki67 of treated and untreated tumors?

As suggested by the reviewer we replaced progression by growth.

Unfortunately, it is impossible at this time to perform the suggested histology analysis in treated or untreated tumors.

Labelled TCA intermediates in figure 2f are difficult to read, please consider to replace with a panel similar to Fig.4a

As suggested by the reviewer, we increased the size of the graphics in the new Fig. 2F.

Corresponding Author Name: FRANCESC VIÑALS
Journal Submitted to: EMBO MOLECULAR MEDICINE
Manuscript Number: EMM-2019-11217-V2